# ContextPRM: Leveraging Contextual Coherence for multi-domain Test-Time Scaling

**Haotian Zhang**[1,2]   **Liu Liu**[1,2*]   **Baosheng Yu**[6]   **Jiayan Qiu**[5]
**Likang Xiao**[2]   **Yanwei Ren**[1,2]   **Quan Chen**[4]   **Xianglong Liu**[3]

[1]School of Artificial Intelligence, Beihang University
[2]Hangzhou International Innovation Institute, Beihang University
[3]School of Computer Science, Beihang University   [4]Kuaishou Technology
[5]University of Leicester   [6]Nanyang Technological University

## ABSTRACT

Process reward models (PRMs) have demonstrated significant efficacy in enhancing the mathematical reasoning capabilities of large language models (LLMs) by leveraging test-time scaling (TTS). However, while most PRMs exhibit substantial gains in mathematical domains, the scarcity of domain-specific training data and knowledge-based learning patterns limits their generalization ability when faced with other domains. To address this limitation, we shift the learning objective from verifying domain-specific knowledge to modeling domain-agnostic logical flow. Centering on *contextual coherence* between chain-of-thought (CoT) steps, our approach is realized through a novel data annotation and training framework, which enhances the model's generalization capabilities across diverse domains. For instance, our resulting model, **ContextPRM**, achieves a notable 6.5% average accuracy improvement over the majority voting baseline via weighted majority voting across nine non-mathematical domains in MMLU-Pro, including law, history, and philosophy, significantly surpassing the 2.2% improvement from VersaPRM and 0.5% gains from other mathematics-focused PRMs, demonstrating consistent performance across both mathematical and non-mathematical domains.

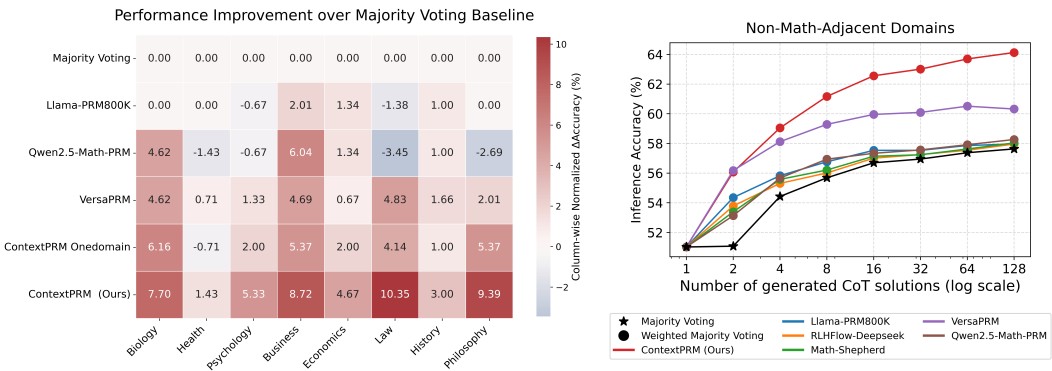

Figure 1: Key results demonstrating ContextPRM's effectiveness. **(Left)**: Training on a single non-math domain surpasses the prior multi-domain SOTA, VersaPRM. **(Right)**: In non-math-adjacent domains, ContextPRM consistently outperforms all baselines under WMV sampling, showcasing superior generalization.

---

*Corresponding author: `liuliubh@buaa.edu.cn`.

Code is available at `https://github.com/shintaro329/ContextPRM`.

# 1 INTRODUCTION

Recent advancements in large language models (LLMs) have fostered significant interest in enhancing their reasoning abilities (Wei et al., 2022; Kojima et al., 2022; Madaan et al., 2023; Chowdhery et al., 2023). Among the strategies explored, test-time scaling (TTS) has emerged as a prominent approach, leveraging additional computational resources during inference to improve performance (Cobbe et al., 2021; Yao et al., 2023a;b; Jaech et al., 2024; Snell et al., 2025). The effectiveness of TTS methods, however, relies heavily on the quality of the underlying verifiers that guide the reasoning process (Lightman et al., 2023). Process Reward Models (PRMs) represent a key class of such verifiers, and have demonstrated strong performance in reranking generated solutions through step-by-step verification (Uesato et al., 2022; Wan et al., 2024). Despite their promise, PRMs face several inherent limitations that restrict their broader applicability.

While several studies demonstrate the strong performance of PRMs on mathematical reasoning tasks (Wang et al., 2024a; Zhang et al., 2025; Yang et al., 2024), most research remains concentrated within the math domain, focusing on synthetic data generation (Lightman et al., 2023; Wang et al., 2024a), training methodologies (Xiong et al., 2024; Zhao et al., 2025), and evaluation benchmarks (Zheng et al., 2024; 2023; Song et al., 2025). Their application to non-mathematical reasoning, however, is still insufficiently explored. VersaPRM (Zeng et al., 2025) highlights this limitation, showing that mathematical PRMs transfer poorly to non-mathematical tasks. It further proposes an efficient and automated method for generating and annotating multi-domain training data, which partially improves PRM performance outside the mathematical domain. However, a critical gap remains in developing a training methodology that enables genuine multi-domain generalization, which requires unifying heterogeneous reasoning styles across disciplines, from the formal and symbol-dense CoTs of the sciences to the nuanced argumentation typical of the humanities.

To address this limitation, we introduce **ContextPRM**, a novel approach that models the *contextual coherence* between steps in a reasoning chain, rather than evaluating them solely for isolated, domain-specific correctness. By shifting the learning objective toward a domain-agnostic logical flow through a novel data annotation standard and context-aware training methodology, ContextPRM explicitly learns to evaluate the logical transitions between consecutive reasoning steps. This design substantially improves cross-domain generalization, establishing a new state-of-the-art in multi-domain test-time scaling. Our main contributions are threefold:

- To better leverage training data and reinforce coherence and logical consistency in chains of thought (CoTs), we introduce the Context-Aware Training Method for PRMs, which combines a new data annotation standard with a context-aware approach to evaluate logical transitions between reasoning steps. e demonstrating that our context-focused method significantly enhances the cross-domain generalizability of PRMs, particularly in domains beyond mathematics.

- We present extensive empirical evidence showing that our context-centered approach substantially improves the cross-domain generalizability of PRMs, achieving notable gains in domains beyond mathematics and demonstrating its versatility and robustness.

- Our resulting model, ContextPRM, achieves a new state-of-the-art for multi-domain process reward modeling, delivering consistent performance improvements across both mathematical and non-mathematical domains, and substantially outperforming prior work. Specifically, in domains non-adjacent to mathematics, ContextPRM achieves a 6.5% average accuracy improvement over the majority voting baseline using Weighted Majority Voting, significantly surpassing the 2.2% gain from the previous state-of-the-art VersaPRM.

# 2 RELATED WORK

## 2.1 PROCESS REWARD MODELS

PRMs outperform Outcome Reward Models (ORMs) in enhancing LLM chain-of-thought reasoning, particularly for mathematical tasks (Li et al., 2023; Zhang et al., 2024b;a; Wu et al., 2025). Recent approaches reduce reliance on costly human annotations through automated supervision, data synthesis, or weak and preference-based signals (Wang et al., 2024a; Lightman et al., 2023; Zeng et al., 2025), while frameworks for PRM training and evaluation improve scalability, robustness, and fine-grained assessment (Yuan et al., 2025; Zhang et al., 2025; Xiong et al., 2024; Zheng et al.,

2024). However, these methods primarily focus on mathematical reasoning (Sprague et al., 2025), and early efforts to generate multi-domain PRM data often lack optimized training strategies (Zeng et al., 2025). In contrast, our approach tailors training to multi-domain data by emphasizing domain-agnostic logical flow, enabling robust cross-domain generalization.

## 2.2 TRAINING METHOD FOR PRMS

As step verifiers for inference LLMs, PRMs are required to assign stepwise reward scores to each reasoning step within a CoT. Conventional training approaches formulate this as a binary classification task, where the PRMs learns to predict correctness labels $\{0, 1\}$ for individual steps. Recent studies have attempted to enhance PRM's performance with various training method improvements. (Zhang et al., 2024a; Zhao et al., 2025; She et al., 2025; Khalifa et al., 2025) first trained the model to generate intermediate reasoning before producing digit rewards, trading increased test time cost for improved accuracy. (Pala et al., 2025) adds additional labels to train PRMs to detect error patterns before reward assignment. However, these methods remain largely constrained to knowledge-intensive mathematical domains. In contrast, our approach introduces a context-aware training methodology tailored for multi-domain data, which explicitly teaches the PRM to evaluate the logical transitions between consecutive reasoning steps.

## 2.3 TEST-TIME SCALING

Reranking-based inference methods, commonly used in test-time scaling for chain-of-thought (CoT) reasoning, select the final answer from a set of sampled CoT solutions using aggregation strategies. Let $a_C$ denote the answer extracted from a CoT solution $C$, and $\mathcal{C}_N = \{C_1, C_2, \ldots, C_N\}$ be $N$ CoTs sampled independently from a generator for a given question. Aggregation functions—such as min-, mean-, and max-aggregation—combine the stepwise scores assigned by the PRM into a single chain-of-thought (CoT) score, which serves as the foundation for downstream reranking (Lightman et al., 2023; Wang et al., 2024a; Sun et al., 2024). Building on this, Majority Voting (MV) (Wang et al., 2023) selects the most frequently occurring answer across sampled CoTs, while Best-of-N (BoN) identifies the answer derived from the CoT with the highest aggregated score. Weighted Majority Voting (WMV) (Uesato et al., 2022) further refines MV by weighting each answer's frequency according to its CoT score, thereby integrating both occurrence and quality signals. Collectively, these methods enhance inference robustness and reliability by exploiting multiple reasoning traces during test-time scaling, ultimately leading to more consistent and higher-quality outputs. This limitation motivates our development of a context-aware approach that explicitly evaluates the logical transitions within CoTs, thereby providing a richer signal for both training and inference.

## 3 METHOD

In this section, we describe the methodology for our context-aware process reward model (ContextPRM). We begin by highlighting the key differences between our training paradigm and conventional PRMs, focusing on how our approach models reasoning coherence. We then present the context-coherence-based CoT annotation pipeline, which captures contextual dependencies within reasoning steps, and detail how ContextPRM is applied during test-time computation.

### 3.1 CONTEXT-AWARE PRMS

Our context-aware PRMs extend conventional PRMs by explicitly considering the relationships between consecutive reasoning steps. While traditional PRMs evaluate the correctness of each step in isolation, context-aware PRMs assess reasoning coherence across the contextual reasoning steps, enabling more accurate and nuanced reward assignment. In the following, we first review conventional PRMs to establish the baseline, and then introduce our context-aware training approach, which leverages contextual dependencies to enhance multi-step reasoning evaluation.

#### 3.1.1 CONVENTIONAL PRMS

Recent works on PRMs learn to assign a scalar reward to each step of a reasoning process. The model is trained as a binary classifier to distinguish between correct and incorrect reasoning steps

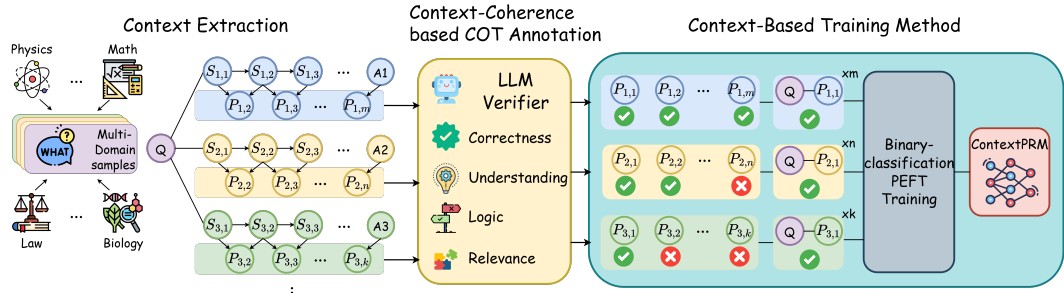

Figure 2: An overview of the ContextPRM training pipeline. Given a multi-domain question $Q$ with multiple CoTs, each CoT is decomposed into a sequence of step pairs $(P_{i,j})$, where $i$ indexes the $i$-th CoT and $j$ indexes the $j$-th step within it. Each pair is annotated for logical coherence using our proposed standard, i.e, correctness, understanding, logic, and relevance. During the context-based training phase, each annotated context pair is combined with the original question $Q$ to construct training samples for binary classification PEFT, ultimately producing the final ContextPRM model.

based on the preceding context (Wang et al., 2024a; Zhang et al., 2025). Formally, for a given question $Q$ and a $k$-step Chain-of-Thought (CoT) denoted as $C = \{S_1, S_2, \ldots, S_k\}$, the training data is structured as a sequence of cumulative prefixes. For each step $S_i$, the model $\mathcal{M}$ receives the concatenated input text $T_i = Q \oplus S_1 \oplus \cdots \oplus S_i$, where $\oplus$ denotes concatenation. Each step $S_i$ is labeled with a binary value $l_i \in \{0, 1\}$, where 1 denotes correct steps and 0 denotes incorrect ones.

This classification is typically achieved by computing a cross-entropy loss over the logits of predefined positive ('+') and negative ('-') tokens. The loss is computed only at the position of the final token of step $S_i$, which we denote by $p_i$. Let the model with parameters $\theta$ output logits $\mathbf{o}_i = \mathcal{M}(T_i; \theta)$. The loss function for the entire CoT is defined as:

$$\mathcal{L}_{\text{PRM}}(\theta) = \sum_{i=1}^{k} \text{CrossEntropy}(\mathbf{o}_{i,p_i}^{\{t_-, t_+\}}, l_i), \tag{1}$$

where $\mathbf{o}_{i,p_i}^{\{t_-, t_+\}}$ represents the logits for the negative token $t_-$ and positive token $t_+$ at position $p_i$.

However, under this training paradigm, as the context length grows with each step, the model struggles to diagnose the true root cause of current step failures. Furthermore, as the context length grows with each step, it becomes increasingly difficult for the model to capture the contextual logic that is most relevant to the correctness of the current step $S_i$, often failing to distinguish between isolated factual errors and deeper contextual fallacies.

### 3.1.2 CONTEXT-AWARE TRAINING

To enhance the utilization of information within training samples and to reinforce the model's understanding of coherence and logical consistency in CoTs, we propose the Context-Aware Training Method for PRM. This approach optimizes the model's context-awareness by constructing a composite contextualized representation for each reasoning step. Specifically, we transform each step $S_i$ into a contextualized representation $P_i$ achieved by a formatting function $\mathcal{F}$.

Formally, for each step $S_i$ in the CoT $C$, we define its new context $\tilde{\mathcal{S}} = \{\tilde{S}_1, \ldots, \tilde{S}_k\}$, where $\tilde{S}_i = Q$ if $i = 1$, and $\tilde{S}_i = S_{i-1}$ if $i > 1$. Then, a concat function is applied for each step and its context, which takes the context and the current step as input and structures them into a new sequence by inserting special tokens that explicitly mark the beginning of the context and the step. The resulting contextualized step is denoted as: $P_i = \mathcal{F}(\tilde{S}_i, S_i)$. This formatting explicitly provides the model with the preceding step as direct context.

This method constructs $k$ distinct and independent training samples from a single $k$-step CoT. For each step $i \in [k]$, the input to the model, $\tilde{T}_i$, is formed by directly concatenating the question $Q$ with the contextualized representation of the current step: $\tilde{T}_i = Q \oplus P_i$. The supervision signal is derived from a parallel set of coherence labels, $c_i \in \{0, 1\}$, where 1 denotes a coherent transition.

Table 1: Comparison of First Incorrect Step Identification between Original and Context-Correctness Based Labeling.

| Origin Label | Total | Same | Earlier Wrong | Later Wrong | Modification rate |
|---|---|---|---|---|---|
| Correct | 36935 | 27823 | 9112 | \ | 24.67% |
| Incorrect | 47163 | 18931 | 26904 | 1328 | 57.04% |
| Total | 84098 | 46754 | **36016** | 1328 | **42.82%** |

The loss is computed at the final token position of the contextualized step $\tilde{S}_i$, denoted by $\tilde{p}_i$. The ContextPRM loss is then calculated by summing the losses for each of these $k$ samples:

$$\mathcal{L}_{\text{ContextPRM}}(\theta) = \sum_{i=1}^{k} \text{CrossEntropy}(\tilde{\mathbf{o}}_{i,\tilde{p}_i}^{\{t_-, t_+\}}, c_i), \tag{2}$$

where the logits $\tilde{\mathbf{o}}_i = \mathcal{M}_{\text{ContextPRM}}(\tilde{T}_i; \theta)$ are produced from the corresponding contextualized input $\tilde{T}_i$. This method compels the model to focus not on the isolated correctness of a step, but on the logical validity of the transition between steps.

A critical component of our methodology is the tight coupling of our training method with its corresponding supervision signal. As our approach aims to shift the learning objective towards contextual coherence, reusing traditional annotation methods and correctness labels ($l_i$) would create a fundamental mismatch between the training objective and the annotation logic. Therefore, to ensure that the training and data labels are properly aligned, we introduce a new annotation standard (detailed in Section 3.2) that is specifically designed to match our context-aware training paradigm, thereby maximizing the effectiveness of our method.

## 3.2 CONTEXT-COHERENCE-BASED CoT ANNOTATION

During the data annotation phase, we proposed an annotation standard to accurately assess each context pair of a CoT. This standard is designed not only to identify factually incorrect steps but also to discern those that are correct in isolation yet are logically flawed or unfounded within the broader context of the reasoning chain. Inspired by the 3-level annotation method in previous works (Lightman et al., 2023; Zeng et al., 2025), we categorize each step in a context pair based on the following criteria:

> **Context-Coherence Based CoT Annotation Criteria**
>
> - **Good:** The step is correct, verifiable, contextually appropriate, and contributes insightfully to the solution path.
> - **Okay:** The step is correct and verifiable but is either redundant or makes only minimal progress toward the solution.
> - **Bad:** The step is categorized as bad if it exhibits one or more of the following flaws.
>   - **Incorrect:** The step contains a factual or calculation error.
>   - **Misinterpretation:** The step is based on a misunderstanding of the problem's premise or the goal.
>   - **Logical Fallacy:** The step contains a structural flaw in its reasoning, such as a non-sequitur or contradiction.
>   - **Misdirection:** The step introduces information or a line of reasoning that is irrelevant to the solution path.

We apply this labeling standard to each context pair within a CoT sequentially until the first 'Bad' step is identified. All subsequent context pairs following this initial error are automatically labeled as incorrect. To quantify the impact of our refined annotation standard, we conducted a comparative analysis against the original labels using `gpt-4o-mini-2024-07-18`, focusing on the identification of the first incorrect step.

The quantitative results of this comparative analysis are presented in Table 1. The table is generated by re-annotating the dataset with our context-coherence criteria and comparing the position of the first identified 'Bad' step against that from the original labels. The CoTs are grouped by their status provided by original labels. We then categorize the comparison into three outcomes: ***Same***, where both methods agree on the position of the first error; ***Earlier Wrong***, where our method identifies the first error at an earlier position; and ***Later Wrong***. The distinction between these outcomes is critical. The ***Earlier Wrong*** category represents the most significant modification, as it provides a fundamentally different and more precise supervision signal by preventing the model from learning from steps built upon a flawed premise. Consequently, we define the ***Modification rate*** as the proportion of CoTs in this category to measure the extent to which our method meaningfully refines the training data.

Notably, our method identifies errors in 24.67% of CoTs previously considered entirely correct, highlighting a stricter standard for logical consistency. For CoTs already containing known errors, our approach locates the initial mistake earlier in a majority of cases (57.04%). The substantial overall modification rate of 42.82% validates that our method provides a significantly different and more precise assessment.

## 4 EXPERIMENTS

### 4.1 EXPERIMENTAL SETUP

**Model Training:** Following the settings in VersaPRM(Zeng et al., 2025), we fine-tuned the open-source base model Llama-PRM800K, which is full-finetuned from Llama-3.1-8B-Instruct with PRM800K dataset. We trained our model with LoRA (Hu et al., 2022) applied to all linear layers with ($r = 16, \alpha = 32$), for 3 epochs with a learning rate of 1e-4 and a total batch size of 32, utilizing our proposed context-based loss function in 3.1.2. To ensure the fairness of comparison, we only updated the label of the training data of VersaPRM with our annotation method as training data, as shown in Table 1. The same hyperparameter configuration is maintained for all other LoRA-tuned models used in our ablation studies. We use 8xRTX 5090 GPUs for model training.

**Evaluation Method:** To assess multi-domain performance, we employ the MMLU-Pro-CoT-Eval (Unlabeled) test set released by VersaPRM, which comprises 2063 questions uniformly distributed across the domains of MMLU-Pro(Wang et al., 2024b), with each question accompanied by 128 candidate CoTs generated by Llama-3.1-8B-Instruct to measure the performance of the PRM when integrated with test-time scaling methods such as Weighted-Majority-Voting (WMV) and Best-of-N (BoN) sampling. We use Min-Aggregation for both methods. To facilitate the comparison of multi-domain capabilities, the test set is further partitioned into a **math-adjacent** split and a **non-math-adjacent** split. The math-adjacent split comprises all questions from the chemistry, computer science, engineering, and physics domains. Conversely, the non-math-adjacent split includes questions from biology, health, psychology, business, economics, law, history, philosophy, and other.

### 4.2 MAIN RESULTS

In main experiments, we compared ContextPRM with several strong open-source, mathematics-focused PRMs, including Qwen2.5-Math-PRM(Zhang et al., 2025), Math-Shepherd(Wang et al., 2024a), RLHFlow-Deepseek(Xiong et al., 2024); the state-of-the-art multi-domain PRM VersaPRM(Zeng et al., 2025), and our base model, Llama-PRM800K. The evaluation results are shown with three scopes: all domains average, mathematics-specific domains, and non-math-adjacent domains. Performance is measured using both Weighted Majority Voting (WMV) and Best-of-N (BoN) sampling methods with Majority Voting as general baseline. As shown in Figure 3, the top and bottom rows illustrate the performance under WMV and BoN sampling, respectively. More detailed results are available in Appendix B.

- **Under WMV sampling:** ContextPRM achieves state-of-the-art results across all domains combined and non-math-adjacent domains. Overall, ContextPRM gains 5.4 accuracy improvement for all-domain questions. In non-math-adjacent domains, ContextPRM improves 6.5% accuracy over the Majority Voting baseline, demonstrating superior multi-domain adaptability. Concurrently, ContextPRM maintains highly competitive performance within the mathematics domain.

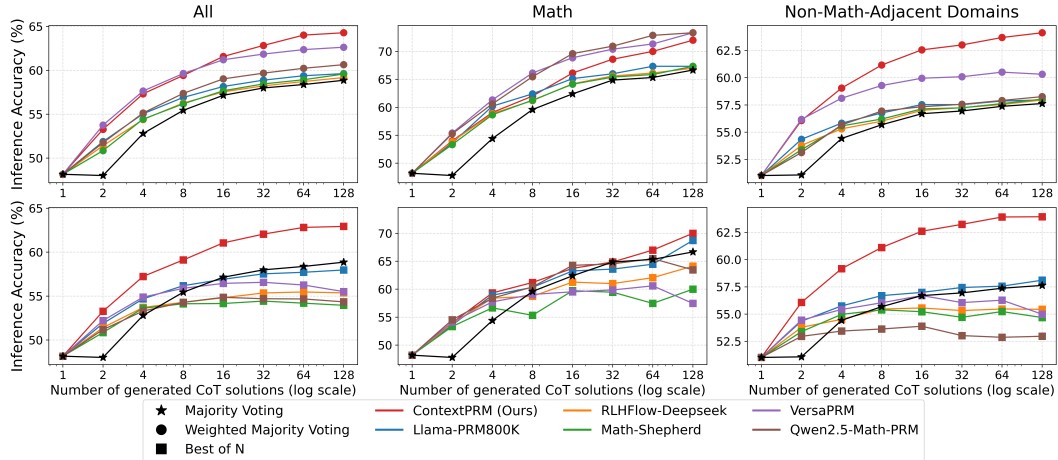

Figure 3: Comparison of WMV (top) and BoN (bottom) under Min-Aggregation using ContextPRM (Ours) against VersaPRM and other open-source math PRMs on MMLU-Pro-CoT-Eval (Unlabeled). ContextPRM outperforms other models in overall results, and the performance gap becomes largest in domains not adjacent to Math, demonstrating its superior generalization.

- **With BoN sampling:** ContextPRM continues to deliver state-of-the-art performance, securing a 6.3% accuracy improvement over the Majority Voting baseline in non-math-adjacent domains, further validating its strong domain generalization capabilities.

**Notation:** Compared to its base model (Llama-PRM800K), the performance of ContextPRM is substantially elevated after LoRA fine-tuning with our context-based training method (Section 3.1.2), which attests to the efficacy of our proposed approach.

## 4.3 ABLATION STUDIES ON CONTEXT METHODS

In this section, we conduct ablation studies to investigate the individual contributions of our proposed context-based training method in 3.1.2 and context-based labeling method in 3.2. As illustrated in Figure 4, we trained two ablation models. The first model"Context-train Only" only employs our context-based training method but utilizes the original, unmodified training data labels, while "Context-label Only" is trained using the data with our updated context-based labels but adheres to the standard PRM training method. For comparison, our full ContextPRM model incorporates both methods, while the VersaPRM baseline employs neither. A summary of the configurations is provided in Table 2. Detailed results are available in Appendix C.

The results reveal a strong synergistic effect between our proposed methods. In the math domain, models trained with either ablation method perform nearly identically to VersaPRM, whereas our full ContextPRM shows a 2.2% performance decrease, suggesting a trade-off for generalization. This trade-off is validated in the non-math-adjacent domains, where the ablation models offer only a marginal improvement over VersaPRM ( 0.5%), but the complete ContextPRM, integrating both methods, achieves a substantial 4.3% increase in accuracy.

**Context-label Only.** When using the standard training method with our improved labels, the model's performance is largely comparable to the VersaPRM baseline, exhibiting parity in the math domain and a very slight improvement of 0.84% in non-math-adjacent domains,which is intentional compared with the modification rate when our annotation method applied, as shown in Table 1. This observation highlights that our annotation standard is primarily designed to rectify logical inconsistencies, rather than to artificially boost training performance by increasing the quality of labels. The result confirms that our main contribution is a more context-coherence aware supervision signal.

**Context-train Only.** Similarly, the comparison between the "Context-train Only" model and VersaPRM affirms that our training method also contributes to performance gains in multi-domain test-time scaling. However, a notable mismatch exists between the training objective and the supervision

Table 2: Configuration of models used in the ablation study.

| Model | Context-based Training | Context-based Labeling |
|---|---|---|
| VersaPRM (Baseline) | X | X |
| Context-train Only | ✓ | X |
| Context-label Only | X | ✓ |
| ContextPRM (Ours) | ✓ | ✓ |

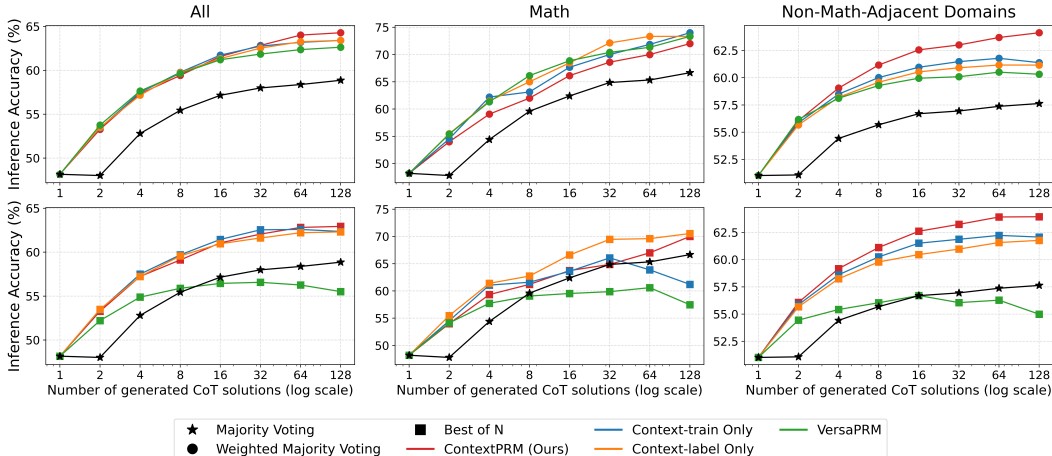

Figure 4: Ablation study results on the contributions of context-based training and context-based labeling. The combined approach (ContextPRM) consistently yields the best performance in non-math-adjacent domains.

signal. Despite this configuration that forces the model to infer contextual relationships from a signal that lacks this explicit information, the model still consistently outperforms the baseline, achieving gains of 0.67% in the math domain and 1.07% in non-math-adjacent domains. This result robustly demonstrates the inherent superiority of our context-based training framework, as it elicits better performance even when supplied with a sub-optimal, non-contextual supervision signal.

**Full Integration (ContextPRM).** The results reveal a strong synergistic effect between our proposed methods. In the math domain, models trained with either ablation method perform nearly identically to VersaPRM. In contrast, the full ContextPRM shows a 2.2% performance decrease, suggesting a trade-off for generalization. This trade-off is further validated in the non-math-adjacent domains where ContextPRM, integrating both methods, achieves a substantial 4.3% increase in accuracy. The trade-off is acceptable as shown in Figure 3, where ContextPRM still significantly outperforms its base model (Llama-PRM800K) in mathematics and also maintains state-of-the-art performance on the combined 'All' domains. This outcome validates the success of shifting the training objective toward context coherence during the PEFT stage and demonstrates the overall effectiveness of our integrated approach.

### 4.4 ABLATION STUDIES ON DOMAIN GENERALIZATION

In this section, we investigate the inter-domain relationships and assess whether our method retains its multi-domain capabilities when trained on a limited, single-domain dataset. To achieve this, we partitioned the training set by category and applied our integrated context-based annotation and training methodology to each subset individually.

As depicted in Figure 5, our method demonstrates remarkable generalization even when fine-tuned on data from certain single domains. Notably, models trained exclusively on domains such as 'Law', 'Psychology', or 'Philosophy' still achieve strong performance in multi-domain evaluations. This result underscores the versatility of our approach, suggesting that by focusing on domain-agnostic

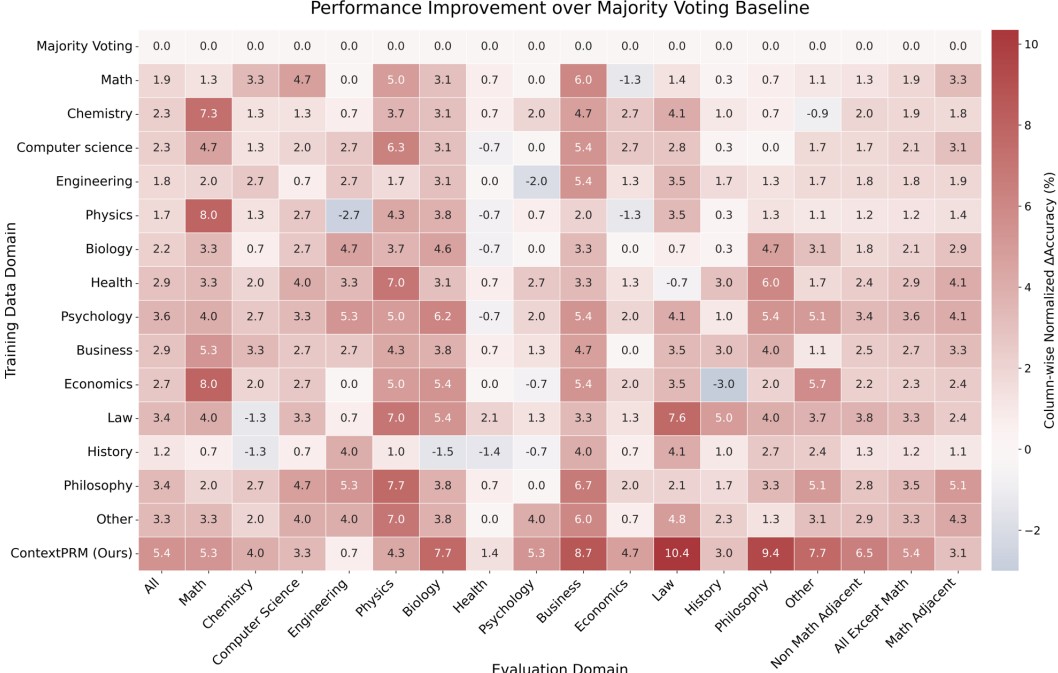

Figure 5: Performance improvement over the Majority Voting baseline. Each row represents a model fine-tuned on a single training domain, while each column represents an evaluation domain. The results highlight the strong cross-domain generalization of models trained with our method.

logical coherence, the model can learn transferable reasoning skills that are effective even beyond the scope of its specific training data. A closer analysis of the heatmap reveals two key insights:

**Training on non-math-adjacent domains yields better multi-domain generalization.** The rows corresponding to models trained on these domains (e.g., 'Psychology', 'Law', 'Philosophy') generally exhibit stronger overall test performance. For our base model trained extensively on mathematics-heavy PRM-800K dataset, these non-math datasets present greater novelty and learning value. They enrich the model's understanding of diverse logical patterns, leading to more significant fine-tuning effects in a sample-efficient setting.

**The effectiveness of fine-tuning is correlated with the logical intensity of the training domain.** Models trained on knowledge-intensive domains such as 'History' (1.2% avg. improvement) and 'Physics' (1.7% avg. improvement) show limited generalization. In contrast, models trained on logic-intensive domains like 'Philosophy' (3.4% avg. improvement), 'Psychology' (3.6% avg. improvement), and 'Health' (2.9% avg. improvement) perform substantially better across the board. This disparity suggests that our method's success is deeply rooted in its ability to leverage the underlying logical structure of CoT data, rather than relying on domain-specific factual knowledge. Notably, despite being trained on data from a single domain, several of these models still surpass the full-dataset performance of VersaPRM (2.7% avg. improvement) in non-math-adjacent domains, demonstrating the data-efficiency of our approach.

## 4.5 ERROR TYPE ANALYSIS

To validate that ContextPRM's gains stem from resolving logical inconsistencies, we classified error types in model failure cases. Analysis reveals that math-intensive domains suffer predominantly from factual "Incorrectness", while non-mathematical domains exhibit a high density of logical errors. By isolating the **Fixed Set** (cases where VersaPRM failed but ContextPRM succeeded), Figure 6 shows ContextPRM primarily rectifies logical errors (e.g., fallacies, misinterpretations), especially in humanities. Furthermore, a correlation analysis (Figure 7) between ContextPRM's performance improvement and the domain's logical error ratio shows a strong positive correlation

($r = 0.80$). This confirms ContextPRM achieves multi-domain generalization specifically by enhancing contextual coherence and correcting logical flaws, rather than refining factual accuracy.

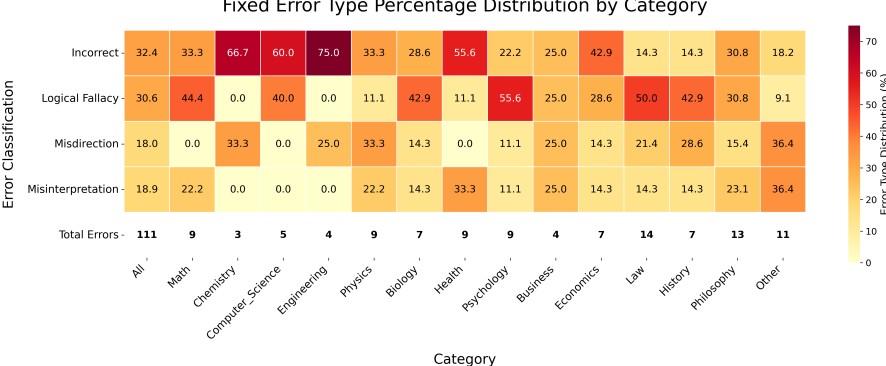

Figure 6: Error type distribution in the Fixed Set. ContextPRM predominantly corrects logical errors.

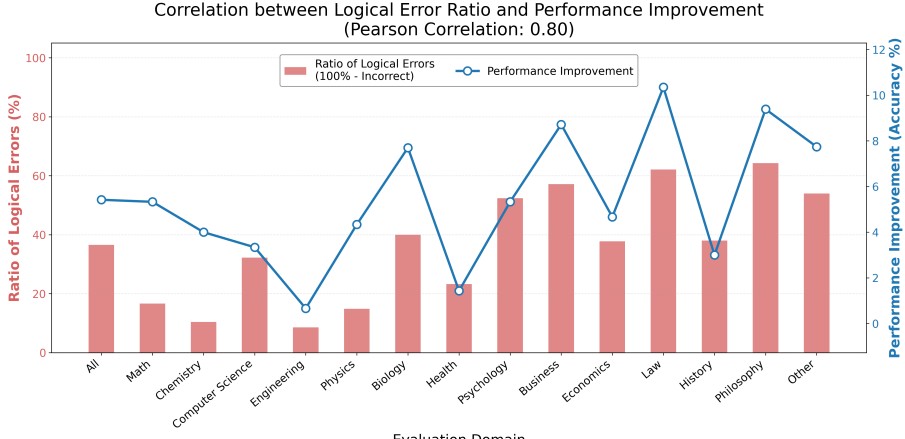

Figure 7: Correlation ($r = 0.80$) between logical error ratio and ContextPRM's performance improvement.

## 5 CONCLUSION

In this work, we addressed the critical challenge of limited cross-domain generalization in Process Reward Models (PRMs). Existing approaches remain constrained by domain-specific correctness signals and thus struggle to accommodate the diverse reasoning styles found across disciplines, from the formal symbolic derivations of the sciences to the nuanced argumentation of the humanities. To overcome this gap, we shifted the learning objective from verifying isolated correctness to modeling domain-agnostic *contextual coherence*. To realize our motivation, we proposed a synergistic framework combining a new data annotation standard with a corresponding context-aware training methodology. Our resulting model, **ContextPRM**, establishes a new state-of-the-art in multi-domain test-time scaling, achieving significant and consistent accuracy improvements across non-mathematical domains where prior models have struggled. Extensive experiments, including ablation and single-domain studies, validate that our approach successfully imbues the model with robust generalization capabilities by teaching it the universal principles of logical flow. Our findings suggest that focusing on the structure of reasoning, rather than its content alone, is a promising direction for building more general and robust process reward models for multi-domain test-time scaling.

## ACKNOWLEDGMENTS

This work was supported by the National Natural Science Foundation of China (Grant No. KZ37132301, KZ37145801, 62441617), Strategic scientific and technological innovation cooperation Project (2025YFE0209100), CCF-Kuaishou Fund for Exploring Large Models (Grant No.CCF-KuaiShou2025012), and the Research Funding of Hangzhou International Innovation Institute of Beihang University (Grant No.2024KQ095),

## STATEMENTS

### ETHICS STATEMENT

The authors of this paper have read and adhered to the ICLR Code of Ethics. Our research focuses on developing Process Reward Models (PRMs) to enhance the logical coherence and reliability of Large Language Models (LLMs), which we believe is a positive contribution. All datasets used are based on the public, non-sensitive MMLU-Pro benchmark, and all base models are used in accordance with their licenses. While we acknowledge the dual-use risk inherent in advancing LLM capabilities, our work is specifically aimed at improving the factuality and logical consistency of model outputs, thereby mitigating risks associated with sophisticated misinformation.

### REPRODUCIBILITY STATEMENT

All details required to reproduce our results are provided in this paper and the forthcoming supplementary materials. Our context-aware training methodology and data annotation standards are described in Section 3. The experimental setup, including hyperparameters and evaluation protocols, is detailed in Section 4. All datasets and baseline models used for comparison are publicly available on HuggingFace, as detailed in Table 3. The source code for our data processing, training, and evaluation pipeline, training data and model checkpoints will be made publicly available upon publication to facilitate further research.

Table 3: Publicly available assets used in our experiments.

| Asset Type | HuggingFace Path |
|---|---|
| *Datasets* | |
| Training Data | UW-Madison-Lee-Lab/MMLU-Pro-CoT-Train-Labeled |
| Evaluation Data | UW-Madison-Lee-Lab/MMLU-Pro-CoT-Eval |
| *Baseline Models* | |
| Math-Shepherd | peiyi9979/math-shepherd-mistral-7b-prm |
| RLHFlow-Deepseek | RLHFlow/Llama3.1-8B-PRM-Deepseek-Data |
| VersaPRM | UW-Madison-Lee-Lab/VersaPRM |
| Llama-PRM800K | UW-Madison-Lee-Lab/Llama-PRM800K |
| Qwen2.5-Math-PRM | Qwen/Qwen2.5-Math-PRM-7B |

### THE USE OF LARGE LANGUAGE MODELS (LLMS)

The authors of this paper have read the The Use of Large Language Models (LLMs) policies. The authors of this paper have confirmed that LLMs were not used for research ideation or for paper writing. All conceptual work, experimental design, data analysis, and manuscript composition were conducted by the authors.

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

# A  ERROR TYPE ANALYSIS

In this section, we conduct a granular analysis of the specific error types manifested in the failure cases of both ContextPRM and the VersaPRM baseline across each subdomain of the test set. Our objective is to empirically validate that the performance gains achieved by ContextPRM stem from its enhanced capacity to identify logical inconsistencies. Furthermore, we provide a correlation analysis to explicitly demonstrate the relationship between the model's ability to detect logical errors and the overall improvement in reasoning accuracy.

## A.1  ERROR TYPE ANALYSIS BY QUESTION CATEGORY

As illustrated in Figure 8, 9 and Figure 10, we conducted a comprehensive error type analysis by extracting all failure instances for both ContextPRM and VersaPRM from the complete test set. Additionally, we measured the Fixed Set, which contains instances where the baseline failed but ContextPRM successfully predicted the correct outcome to investigate the nature of the improvements. To ensure consistency, we employed `gpt-4o-mini-2024-07-18` to classify the error types of failure cases, strictly adhering to the same classification taxonomy defined in Section 3.2.The visualized data reveals distinct trends:

- **Dominance of Factual Errors in Math-Adjacent Domains.** In mathematically intensive domains such as Mathematics and Physics, the proportion of the Incorrect error type (factual calculation or definition errors) is markedly higher compared to humanities disciplines. While solving problems in these domains necessitates rigorous logic, the correctness of a Chain-of-Thought (CoT) is predominantly governed by the self-consistency and absolute accuracy of high-frequency definitions, formulas, and symbolic manipulations, which are sparse in non-mathematical domains, leading to a different error distribution.

- **Reduction of Logical Errors in ContextPRM.** Within the failure cases of VersaPRM, the proportion of logical errors ((Logical Fallacy, Misinterpretation, and Misdirection) is significantly higher than that observed in ContextPRM. This discrepancy indicates that ContextPRM possesses a superior capability to detect and penalize logical flaws. Consequently, this specific capability directly contributes to the reduction in the total number of errors, validating the effectiveness of our context-aware training objective in filtering out logically incoherent reasoning paths.

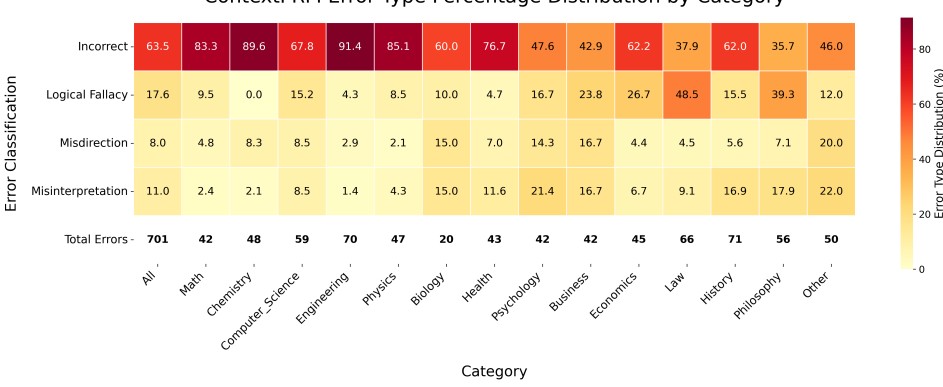

Figure 8: Distribution of error types in failure cases for ContextPRM across different domains. The model demonstrates a lower prevalence of purely logical errors compared to the baseline.

- **ContextPRM Improves Reasoning Primarily by Correcting Logical Errors.** The analysis on Fixed Set (Figure 10) demonstrates that the majority of errors rectified by ContextPRM belong to logical categories, rather than simple factual inaccuracies (Incorrect). Notably, this trend is even more pronounced in humanities and non-math-adjacent domains, where logical errors occur more frequently than calculation errors. This confirms that ContextPRM enhances reasoning capabilities primarily by identifying and correcting flaws in

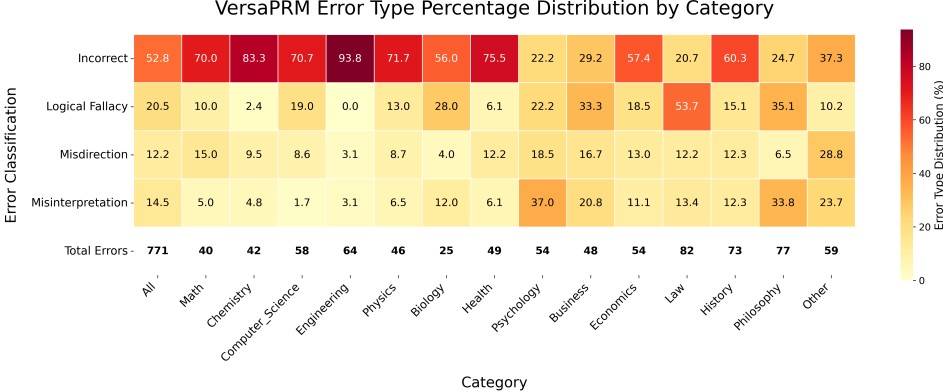

Figure 9: Distribution of error types in failure cases for VersaPRM (Baseline). A higher proportion of logical fallacies and inconsistencies is observed compared to ContextPRM.

logical flow and contextual coherence, effectively mitigating the non-sequiturs and inconsistencies that often plague traditional reward models in these complex, qualitative domains.

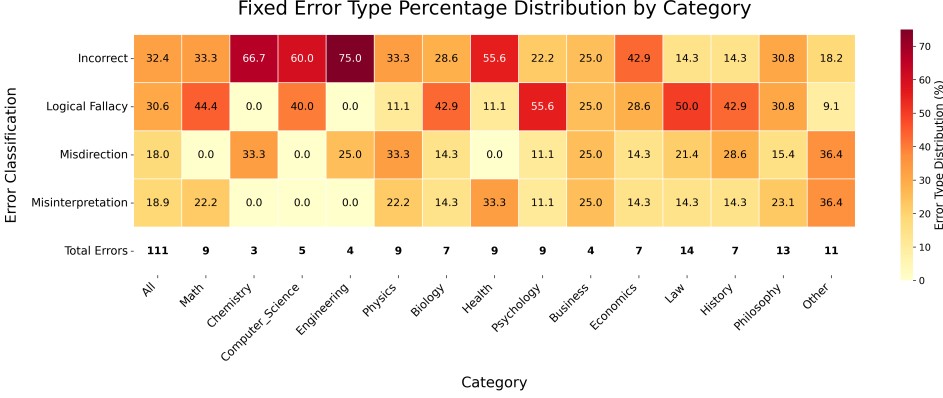

Figure 10: Distribution of error types in the Fixed Set (samples where VersaPRM failed but ContextPRM succeeded). The heatmap highlights that ContextPRM predominantly corrects logical errors.

## A.2 CORRELATION ANALYSIS

To provide a quantitative assessment of whether the efficacy of ContextPRM is indeed driven by its handling of logical errors, we conducted a correlation analysis between two key variables across different domains: the *Performance Improvement* achieved by ContextPRM over the baseline, and the *Ratio of Logical Errors in Majorrity Voting Baseline* of each question category (calculated as $100\% -$ Percentage of Incorrectness Errors).

As depicted in Figure 11, we observe a strong positive correlation between the prevalence of logical errors in a domain and the magnitude of the performance gain yielded by ContextPRM. Domains with a higher density of logical pitfalls, such as Philosophy and Law, exhibit the most substantial performance boosts. This strong statistical relationship provides compelling empirical evidence that ContextPRM realizes its improvements specifically by correcting logical flaws and enhancing contextual coherence, rather than merely refining factual accuracy.

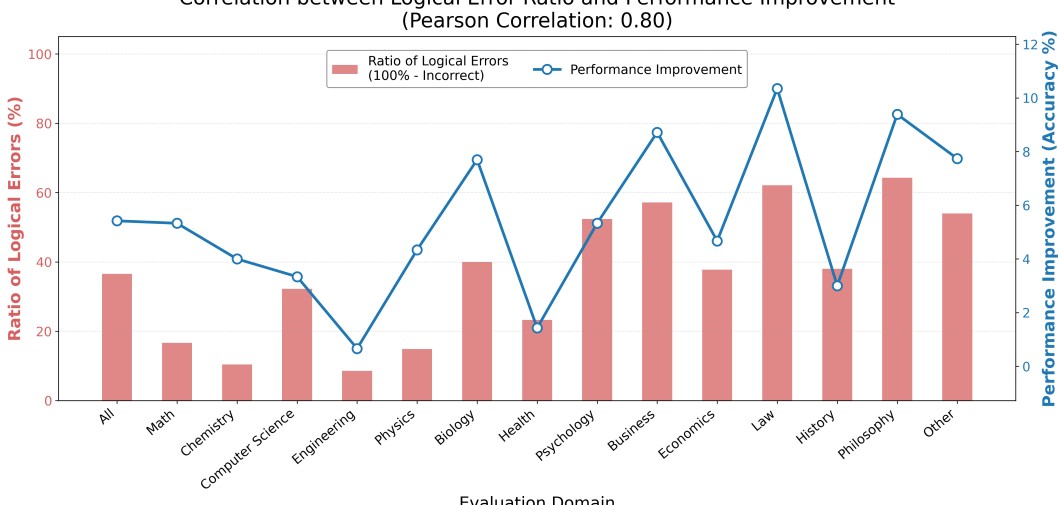

Figure 11: Correlation analysis between the ratio of logical errors in a domain and the performance improvement achieved by ContextPRM. The strong positive correlation ($r = 0.80$) confirms that our method is most effective in domains where reasoning failures are primarily driven by logical inconsistencies rather than factual inaccuracies.

## B    DETAILED RESULTS FOR MAIN EXPERIMENT

In this section, we provide a comprehensive, domain-wise breakdown of our main experimental results. These figures supplement the aggregated results presented in Section 4.2 and offer a more granular view of model performance across the diverse disciplines within the MMLU-Pro-CoT-Eval benchmark. For each of the two primary test-time scaling methods, Weighted Majority Voting (WMV) and Best-of-N (BoN) sampling, we present both a line plot illustrating performance as a function of the number of candidate CoTs and a heatmap visualizing the performance improvement over the majority voting baseline.

### B.1    DETAILED RESULTS FOR WMV SAMPLING

Figure 12 displays the inference accuracy of all evaluated models under the WMV sampling method across each individual domain. The horizontal axis represents the number of generated CoT solutions on a logarithmic scale, while the vertical axis shows the resulting accuracy. Figure 13 provides a complementary view, showing the column-wise normalized accuracy improvement of each model over the standard majority voting baseline.

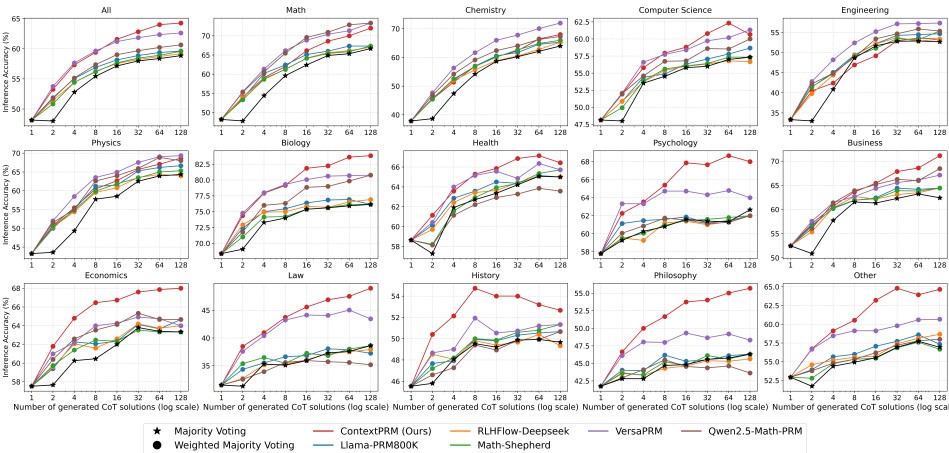

Figure 12: Detailed domain-by-domain performance comparison under WMV sampling. ContextPRM consistently demonstrates strong performance, particularly in non-math-adjacent domains.

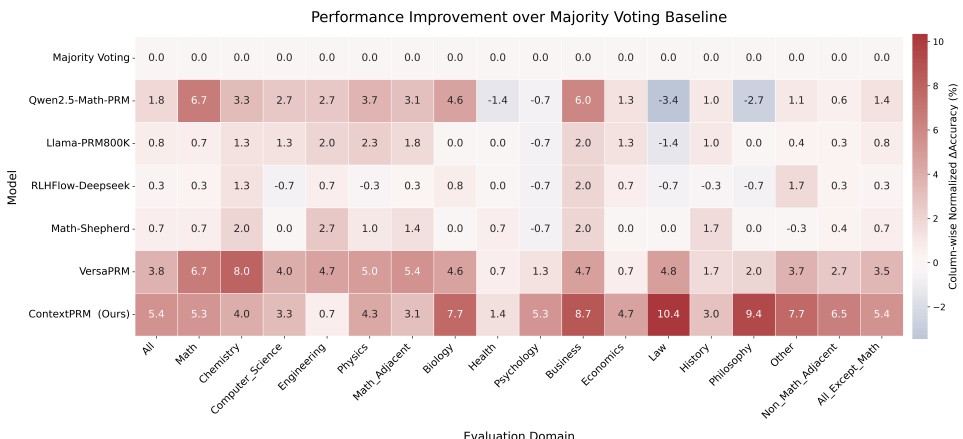

Figure 13: Heatmap of performance improvement over the majority voting baseline with WMV sampling. ContextPRM (bottom row) achieves the most significant and consistent improvements across nearly all evaluation domains, especially in non-mathematical subjects.

## B.2   Detailed Results for BoN Sampling

Similarly, this subsection details the results for the Best-of-N (BoN) sampling method. Figure 14 illustrates the inference accuracy for each model as the number of candidate solutions increases. Figure 15 presents the corresponding heatmap of performance improvements over the majority voting baseline, using a purple-green color scale to visualize the results. These figures confirm that the strong cross-domain generalization of ContextPRM observed under WMV sampling is robust and holds under the BoN evaluation scheme as well.

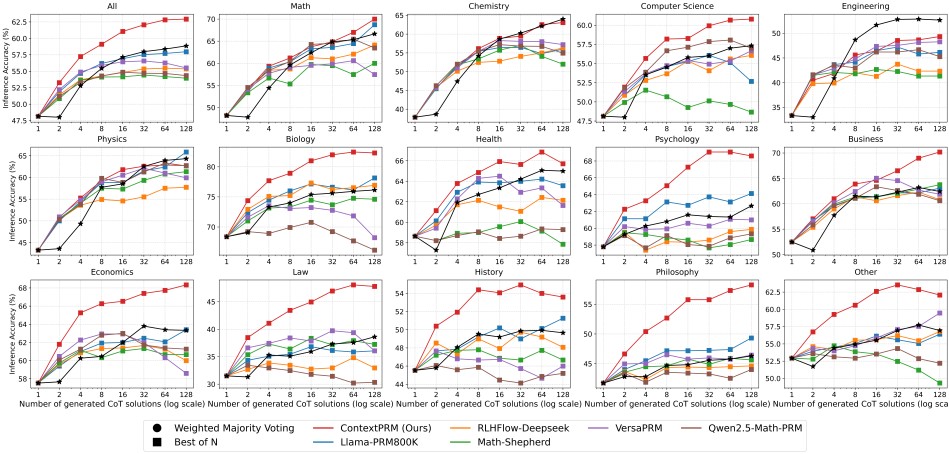

Figure 14: Detailed domain-by-domain performance comparison under BoN sampling. The trend of ContextPRM's superior performance in non-math-adjacent domains continues to hold.

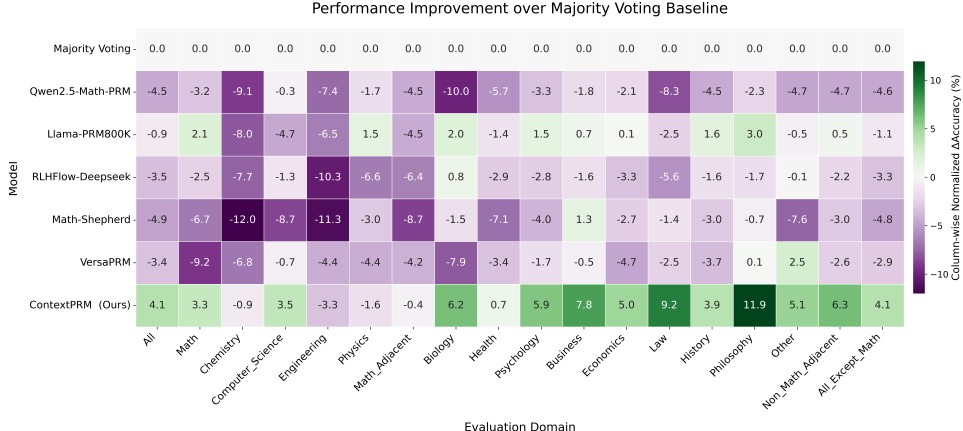

Figure 15: Heatmap of performance improvement over the majority voting baseline with BoN sampling. ContextPRM (bottom row) again shows substantial gains, particularly in logic-intensive domains such as 'Philosophy' and 'History', reinforcing its strong generalization capabilities.

# C DETAILED RESULTS FOR ABLATION ON CONTEXT METHODS

To further dissect the contributions of our proposed components, this section delves into the specifics of the ablation studies referenced in Section 4.3. We present a granular visualization of the performance of models trained with only our Context-Based Training or our Context-Coherence Based Annotation. The following figures illuminate the interplay between these two components under both WMV and BoN sampling schemes, offering a detailed view of their synergistic impact.

## C.1 DETAILED RESULTS FOR WMV SAMPLING

Here, we present the results under the WMV sampling scheme. The line plots in Figure 16 track inference accuracy across domains, while the heatmap in Figure 17 quantifies the improvements over the majority voting baseline. Together, these figures offer a detailed view into the performance trade-offs observed when applying our methods in isolation compared to their full integration.

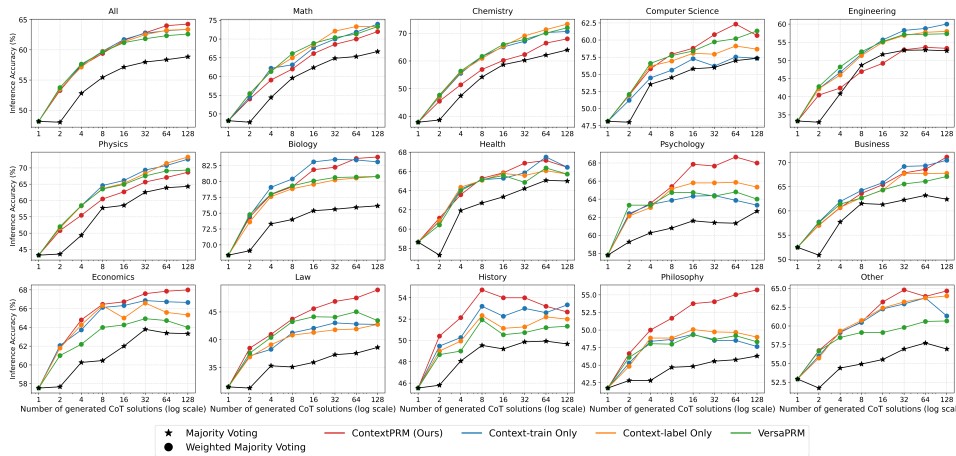

Figure 16: Detailed domain-by-domain performance of ablation models under WMV sampling.

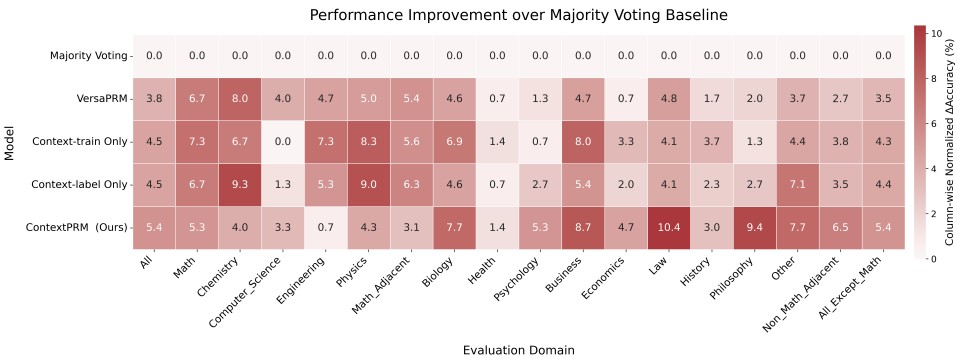

Figure 17: Heatmap of performance improvement for ablation models with WMV sampling.

## C.2 Detailed Results for BoN Sampling

To confirm the robustness of our findings, this subsection provides the corresponding evaluation under the BoN sampling method. The accuracy trends are plotted in Figure 18, with the performance gains visualized in the heatmap in Figure 19. These results reinforce the conclusions drawn from the WMV analysis, showcasing the consistent synergistic relationship between our proposed training and annotation methods.

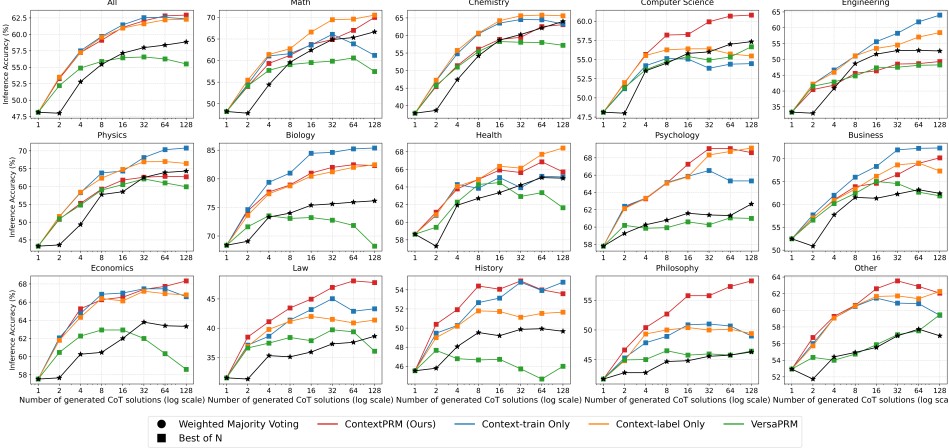

Figure 18: Detailed domain-by-domain performance of ablation models under BoN sampling.

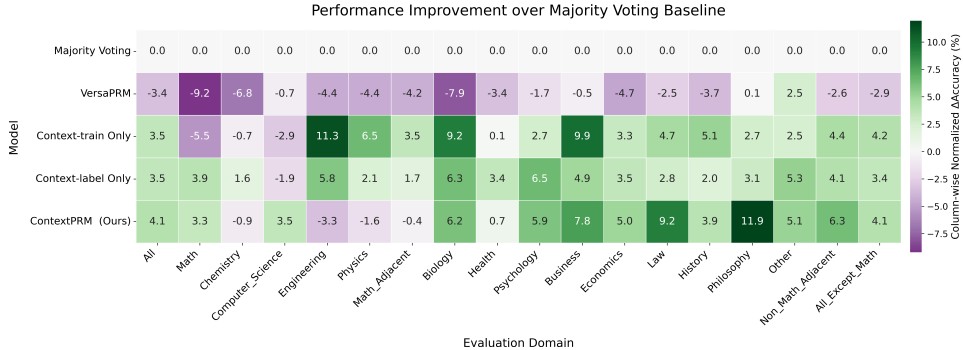

Figure 19: Heatmap of performance improvement for ablation models with BoN sampling.

## D Qualitative Analysis on Selected Examples

In this section, we present a qualitative analysis of selected examples to provide deeper insights into the performance of ContextPRM compared to the VersaPRM baseline. We analyze two key scenarios presented in the following subsections:

**Success Cases** (Subsection D.1), where ContextPRM correctly identifies the first logical error in a reasoning chain that VersaPRM fails to detect. These examples highlight the nuanced understanding of our model.

**Failure Cases** (Subsection D.2), where ContextPRM identifies an error *later* than its actual occurrence according to the ground truth. These examples are chosen to illustrate specific limitations and areas for future improvement.

Each case is presented in a consistent format for clarity. We provide the full context, including the question, options, and the correct answer. For each Chain-of-Thought (CoT) solution, the ground-truth first erroneous step is highlighted in **bold red** for immediate identification. Below the reasoning steps, we display the sequence of rewards assigned by both ContextPRM and VersaPRM, followed by **Explanations** that analyzes the models' performance on that specific case.

## D.1 SUCCESS CASES: CONTEXTPRM CORRECTLY IDENTIFIES ERRORS MISSED BY VERSAPRM

**Question ID**: 7251_mmlu_economics
**Category**: Economics

> **Question:**
>
> Which of the following criticisms of standard ("plain vanilla") GARCH models can be overcome by EGARCH models?
> i) Estimated coefficient values from GARCH models may be negative
> ii) GARCH models cannot account for leverage effects
> iii) The responsiveness of future volatility to positive and negative shocks is symmetric under a GARCH formulation
> iv) GARCH models cannot allow for a feedback from the volatility to the returns
> **Options: A.** (iii) and (iv) only **B.** (i), (iii), and (iv) only **C.** (i) and (iii) only **D.** (ii), (iii), and (iv) only **E.** (i), (ii), and (iii) only **F.** (ii) and (iv) only **G.** (ii) and (iii) only **H.** (i), (ii), (iii), and (iv) **I.** (i) and (iv) only **J.** (i) and (ii) only
> **Answer: E**

**CoT 1:**

> 1. Let's break down the criticisms of standard GARCH models and how EGARCH models address them:
>
> 2. **i) Estimated coefficient values from GARCH models may be negative: This can occur due to the mis-specification of the model. However, it's not a direct criticism of GARCH models that EGARCH models address.**
>
> 3. ii) GARCH models cannot account for leverage effects: Leverage effects refer to the phenomenon where large negative returns are associated with higher volatility. EGARCH models can address this issue by incorporating an asymmetric effect of returns on volatility.
>
> 4. iii) The responsiveness of future volatility to positive and negative shocks is symmetric under a GARCH formulation: This is a criticism of GARCH models. EGARCH models can address this issue by allowing for asymmetric effects of positive and negative shocks on volatility.
>
> 5. iv) GARCH models cannot allow for a feedback from the volatility to the returns: This is not a criticism of GARCH models, but rather a characteristic of the model. GARCH models can allow for feedback from volatility to returns.
>
> 6. Therefore, the correct answer is the one that addresses criticisms (ii) and (iii), which are the leverage effects and symmetry of responsiveness.
>
> 7. The answer is (G).
>
> **ContextPRM rewards**: 1.0000, 0.0173, 0.9961, 1.0000, 0.0090, 0.6758, 1.0000
> **VersaPRM rewards**: 1.0000, 0.9336, 0.9961, 0.9922, 0.7656, 0.8203, 0.8984
> **Explanation**: The VersaPRM fails to identify that the student incorrectly states EGARCH models do not overcome the criticism of potentially negative GARCH coefficient values.

**Question ID**: 6191_mmlu_health
**Category**: Health

> **Question:**
>
> Which of the following vitamins is required for fatty acid b-oxidation?
> **Options: A.** Vitamin C **B.** Vitamin E **C.** Vitamin B12 **D.** Thiamin **E.** Riboflavin **F.** Folic Acid **G.** Vitamin D **H.** Vitamin K **I.** Biotin
> **Answer: E**

**CoT 1:**

1. Fatty acid b-oxidation is a process by which fatty acids are broken down into acetyl-CoA units for energy production.

2. **Biotin is a crucial cofactor for enzymes that catalyze the carboxylation reactions involved in fatty acid b-oxidation.**

3. Therefore, biotin is essential for the process of fatty acid b-oxidation.

4. The answer is (I).

**ContextPRM rewards**: 1.0000, 0.0107, 1.0000, 0.3418
**VersaPRM rewards**: 1.0000, 0.9844, 1.0000, 1.0000
**Explanation**: The VersaPRM fails to detect the incorrect assertion that biotin is a crucial cofactor for fatty acid b-oxidation.

**CoT 2:**

1. Fatty acid beta-oxidation is a process that involves the breakdown of fatty acids to produce energy.

2. **Biotin is a coenzyme that plays a role in fatty acid metabolism and is specifically required for fatty acid beta-oxidation.**

3. The answer is (I).

**ContextPRM rewards**: 1.0000, 0.0025, 0.1099
**VersaPRM rewards**: 1.0000, 0.9922, 1.0000
**Explanation**: The VersaPRM fails to detect the factual error stating Biotin is required for fatty acid beta-oxidation.

**Question ID**: 3393_mmlu_biology
**Category**: Biology

> **Question:**
>
> Which of the following characteristics is predicted for an early-successional plant community?
> **Options: A.** High detrital biomass **B.** High number of predatory species **C.** High presence of fully matured plant species **D.** High frequency of R-selected species **E.** High rates of soil nutrient depletion **F.** High diversity of plant species **G.** High ratios of primary production to standing-crop biomass **H.** High frequency of K-selected species **I.** Low rates of photosynthesis **J.** High niche divergence among co-occurring species
> **Answer: G**

**CoT 1:**

1. Early-successional plant communities are typically characterized by a high abundance of R-selected species.

2. **These species are adapted to rapidly colonize and exploit resources after disturbances such as fire or clear-cutting, but they typically have low levels of reproduction and high levels of mortality.**

3. R-selected species are able to quickly grow and produce large amounts of seeds, which allows them to quickly colonize a new area.

4. These species are usually annual or biennial plants, and they have a high reproductive rate.

5. K-selected species, on the other hand, are adapted to slowly colonize and exploit resources, and they typically have high levels of reproduction and low levels of mortality.

6. They are usually perennial plants that have a high level of competition for resources.

7. In early-successional plant communities, soil nutrient levels are typically high due to the rapid decomposition of organic matter.

8. Therefore, we can expect a high frequency of R-selected species in early-successional plant communities.

9. The answer is (D).

**ContextPRM rewards**: 1.0000, 0.1069, 1.0000, 1.0000, 0.9922, 1.0000, 0.9961, 1.0000, 1.0000

**VersaPRM rewards**: 1.0000, 0.9805, 0.9805, 1.0000, 0.9531, 0.9922, 0.9062, 1.0000, 0.9961

**Explanation**: The VersaPRM fails to detect the factual error regarding R-selected species' reproductive rates in Step 1.

**CoT 2:**

1. **Early-successional plant communities are characterized by a low number of plant species.**

2. They also have a high abundance of fast-growing and highly reproductive species, known as R-selected species.

3. These species have traits that allow them to quickly colonize and exploit available resources.

4. R-selected species are typically small, produce many seeds, and have a short lifespan.

5. In contrast, K-selected species are larger, produce fewer seeds, and have a longer lifespan.

6. The answer is (D).

**ContextPRM rewards**: 0.0240, 1.0000, 1.0000, 1.0000, 1.0000, 1.0000

**VersaPRM rewards**: 0.8203, 1.0000, 1.0000, 1.0000, 1.0000, 1.0000

**Explanation**: The VersaPRM fails to identify that the initial step, though factually correct, misdirects the reasoning from the question's core concept.

**Question ID**: 10995_mmlu_philosophy
**Category**: Philosophy

**Question:**

Shapiro claims that surveys have been confirmed by longitudinal studies that indicate that
**Options: A.** the use of illegal drugs is usually a temporary phase in adolescence. **B.** most individuals experiment with illegal drugs but do not continue use. **C.** excessive use of legal drugs is the norm. **D.** an individual's illegal drug use inevitably diminishes over time. **E.** those who begin using illegal drugs typically stop before addiction sets in. **F.** longitudinal studies have shown no consistent patterns in illegal drug use. **G.** illegal drug users typically transition to legal substances over time. **H.** moderate and/or controlled use of illegal drugs is the norm. **I.** those who begin using illegal drugs typically do not stop until they become addicted. **J.** excessive use of illegal drugs is the norm.
**Answer: H**

**CoT 1:**

1. **The statement that surveys have been confirmed by longitudinal studies is that most individuals experiment with illegal drugs but do not continue use.**

2. This choice is (B).

3. The answer is (B).

**ContextPRM rewards**: 0.0415, 0.9922, 1.0000
**VersaPRM rewards**: 0.8750, 0.9531, 0.9844
**Explanation**: The VersaPRM fails to detect that the student misidentifies the correct claim confirmed by longitudinal studies, selecting option B instead of H.

**CoT 2:**

1. **Shapiro claims that surveys have been confirmed by longitudinal studies that indicate that most individuals experiment with illegal drugs but do not continue use.**

2. This matches option (B), which states that "most individuals experiment with illegal drugs but do not continue use".

3. The answer is (B).

**ContextPRM rewards**: 0.1143, 0.9844, 0.9961
**VersaPRM rewards**: 0.8281, 0.9727, 0.9727
**Explanation**: The VersaPRM fails to detect that the student's initial premise misrepresents Shapiro's claim.

**Question ID**: 5860_mmlu_other
**Category**: Other

> **Question:**
>
> All of the following statements about gerrymandering are true EXCEPT that it
> **Options: A.** is a practice that is universally disallowed in all democratic nations **B.** is a process that is only possible within a multi-party system **C.** is a strategy utilized by both major political parties in the United States **D.** can lead to an increase in voter turnout **E.** can divide like-minded voters into several districts to reduce their influence **F.** is a strategy used to increase the number of districts with a majority of minority voters **G.** has been allowed by the Supreme Court in some instances **H.** increases the control voters have over their elected officials **I.** favors incumbent candidates over challengers
> **Answer: H**

**CoT 1:**

1. Gerrymandering is a practice where electoral districts are manipulated to favor one political party or group over another, often by dividing or packing voters into districts.

2. **Statement D, "can lead to an increase in voter turnout", is not true. Gerrymandering often has the opposite effect, as it can disenfranchise or dilute the voting power of certain groups, leading to decreased voter turnout.**

3. The answer is (D).

**ContextPRM rewards**: 1.0000, 0.4766, 0.0079
**VersaPRM rewards**: 1.0000, 0.9414, 0.8164
**Explanation**: The VersaPRM fails to identify the student's misinterpretation of the problem-solving strategy, specifically the lack of comprehensive evaluation of all options in a 'true EXCEPT' question.

## D.2 FAILURE CASES: CONTEXTPRM IDENTIFIES ERRORS LATER THAN GROUND TRUTH

**Question ID**: 1968_mmlu_law
**Category**: Law

> **Question:**
>
> One Sunday afternoon, a sports fan went to a professional football game. While standing in line at a concession stand, he got into an argument with three men ahead of him in the line. The three men were friends attending the game together. One of the men punched the sports fan in the mouth. A fight ensued. During the fracas, one of the other three men hit the sports fan over the head with a bottle, which caused him to suffer a serious concussion. The sports fan does not know for certain which of the three men struck him with the bottle. If the sports fan asserts a claim against the man who punched him in the mouth to recover damages for the head injury, will the sports fan prevail?
>
> **Options: A.** No, because the sports fan should not have argued with the three men in the first place. **B.** No, because the sports fan does not have a legal basis to sue the man who punched him in the mouth. **C.** No, because the man who punched him in the mouth did not actually strike the sports fan with the bottle. **D.** No, because the sports fan could have avoided the fight. **E.** Yes, because the man who punched him in the mouth was the instigator who struck the first blow that started the fracas. **F.** Yes, because the sports fan was defending himself from an unprovoked attack. **G.** Yes, because the man who punched him in the mouth and the other men were acting in concert. **H.** No, because the sports fan cannot offer proof as to which of the men struck the sports fan over the head with the bottle. **I.** Yes, because the three men were acting as a group and are therefore jointly responsible. **J.** Yes, because all the men were involved in the fight and are therefore liable.
>
> **Answer: G**

**CoT 1:**

> 1. The question is asking whether the sports fan can recover damages from the man who punched him in the mouth.
>
> 2. **Since the sports fan does not know for certain which of the three men struck him with the bottle, he cannot prove that the man who punched him in the mouth was the one who actually caused the head injury.**
>
> 3. Therefore, the sports fan cannot recover damages from the man who punched him in the mouth.
>
> 4. The answer is (H).
>
> **ContextPRM rewards**: 1.0000, 0.9766, 0.4922, 0.1143
> **VersaPRM rewards**: 1.0000, 0.9844, 0.8906, 0.9297
> **Explanation**: While VersaPRM failed to identify any errors, and ContextPRM incorrectly identified Step 2 as the first error, both PRMs critically missed the fundamental legal misinterpretation in Step 1. Step 1 incorrectly assumes that direct causation by the specific defendant is the sole basis for liability, failing to account for principles like 'acting in concert' or joint and several liability. This suggests a shared weakness in both PRMs regarding their understanding and application of complex legal doctrines in tort law.

**CoT 2:**

> 1. **The man who punched the sports fan in the mouth may be liable for the head injury caused by the bottle, but the sports fan cannot prevail because he cannot prove which of the three men actually struck him with the bottle.**
>
> 2. This is an example of the requirement of proximate cause in tort law, which holds that a defendant can only be liable for injuries that are a direct result of their actions.
>
> 3. The answer is (H).
>
> **ContextPRM rewards**: 0.8828, 0.2559, 1.0000
> **VersaPRM rewards**: 0.9062, 0.9062, 0.9336
> **Explanation**: VersaPRM failed to identify any errors in the provided solution. Both ContextPRM and VersaPRM critically missed the fundamental error in Step 0, which incorrectly

concludes that the sports fan cannot prevail due to inability to identify the specific perpetrator of the bottle attack, thereby misinterpreting the legal principle of 'acting in concert' or joint liability. ContextPRM incorrectly identified Step 1 as the first error. This suggests a shared weakness in both PRMs regarding their understanding and application of complex legal doctrines, specifically joint liability and causation in tort law.

**Question ID**: 1986_mmlu_psychology
**Category**: Psychology

### Question:

A person who received a Level 4 rating on the Rancho Los Amigos Scale of Cognitive Functioning:
**Options: A.** is unresponsive to all stimuli but shows signs of basic physiological functioning such as breathing and heartbeat. **B.** is highly responsive and coherent, but cannot remember or integrate past and recent events. **C.** is functioning at an intellectual level that is average for his/her age, education, and demographic background. **D.** is responsive to stimuli but cannot remember or integrate past and recent events, and exhibits impaired judgment. **E.** is confused and incoherent, may exhibit bizarre behavior, and is unable to care for him/herself. **F.** is functioning at an intellectual level that is superior for his/her age, education, and demographic background. **G.** is nonresponsive to visual or auditory stimuli and seems to be in a state of deep sleep. **H.** is alert and oriented and can remember and integrate remote and recent events but may have some impairment in judgment, planning, and abstract reasoning.
**Answer: E**

**CoT 1:**

1. Rancho Los Amigos Scale of Cognitive Functioning categorizes cognitive functioning into eight levels.
2. Level 1: No response - coma
3. Level 2: Generalized response - awake but unable to respond
4. **Level 3: Confused-agitated - confused and incoherent, may exhibit bizarre behavior, and is unable to care for him/herself**
5. Level 4: Confused - responsive to stimuli but unable to remember or integrate past and recent events
6. Level 5: No problem - functioning at an intellectual level that is average for his/her age, education, and demographic background
7. Level 6: Automatic - functioning at an intellectual level that is superior for his/her age, education, and demographic background
8. Level 7: Improvement - continues to improve in functioning, but still impaired in some areas
9. Level 8: Normal - fully recovered and functioning at pre-injury level.
10. Therefore, the description in choice D best matches a person who received a Level 4 rating on the Rancho Los Amigos Scale of Cognitive Functioning.
11. The answer is (D).

**ContextPRM rewards**: 1.0000, 0.9688, 0.9062, 0.9922, 0.1895, 0.8359, 0.7266, 0.8359, 1.0000, 0.9961, 1.0000
**VersaPRM rewards**: 1.0000, 0.9922, 0.9766, 0.9844, 0.9062, 0.8242, 0.8633, 0.9688, 0.9961, 0.8047, 0.9805
**Explanation**: VersaPRM failed to identify any errors in the solution. Both ContextPRM and VersaPRM critically missed the first error in Step 3, which incorrectly describes Level 3 of the Rancho Los Amigos Scale with characteristics that actually belong to Level 4. This indicates a shared weakness in their precise domain knowledge regarding the Rancho Los Amigos Scale.

**Question ID**: 2618_mmlu_psychology
**Category**: Psychology

> ### Question:
>
> The primary advantage of the forced-choice distribution of rated attributes in performance appraisal is that it
> **Options: A.** allows for more flexibility in rating employees **B.** reduces the bias in ratings as it is based on direct observation **C.** is less time-consuming, since it does not require in-depth analysis **D.** is more accurate as it focuses on measurable attributes **E.** is more objective, since it does not deal with the strengths and weaknesses of the employee **F.** enhances the reliability of ratings **G.** eliminates statistical concerns about ipsative measurement **H.** is easier and faster to use, since only two choices are available **I.** eliminates the need for continuous feedback and follow-up **J.** is more comprehensive as it covers all aspects of performance
> **Answer: F**

**CoT 1:**

> 1. **Forced-choice distribution eliminates statistical concerns about ipsative measurement.**
>
> 2. Ipsative measurement refers to the comparison of an individual's performance on a single task or set of tasks across multiple assessments.
>
> 3. Forced-choice distribution eliminates this problem by comparing each rating choice against the other two options.
>
> 4. The answer is (G).
>
> **ContextPRM rewards**: 0.9922, 0.1406, 0.0255, 1.0000
> **VersaPRM rewards**: 0.9961, 0.9805, 0.7812, 0.8672
> **Explanation**: While VersaPRM failed to detect any errors, and ContextPRM incorrectly identified Step 1 as the first error, both PRMs critically missed the fundamental factual error in Step 0. The ground truth states that forced-choice methods often generate ipsative data, which inherently has statistical concerns, rather than eliminating them. This indicates a shared weakness in both PRMs' ability to evaluate the factual correctness of initial premises within this domain.

**CoT 2:**

> 1. **Forced-choice rating involves evaluating two choices, "above average" and "below average", for each attribute.**
>
> 2. This method eliminates the halo effect and leniency, providing a more objective evaluation.
>
> 3. The answer is (G).
>
> **ContextPRM rewards**: 0.9883, 0.2451, 0.7617
> **VersaPRM rewards**: 0.9414, 0.9258, 0.8320
> **Explanation**: While VersaPRM failed to detect any errors, and ContextPRM incorrectly identified Step 1 as the first error, neither PRM recognized the fundamental misinterpretation of 'forced-choice rating' in Step 0. The ground truth clearly states that Step 0's description of the method is an oversimplification and misrepresentation, as it typically involves selecting statements from groups rather than simply evaluating 'above average' and 'below average'. This suggests a shared weakness in their ability to identify foundational conceptual errors related to specific performance appraisal methodologies.

