# OpenReview forum: "ContextPRM: Leveraging Contextual Coherence for multi-domain Test-Time Scaling"
_ICLR.cc/2026/Conference — ICLR 2026 Poster_

### Official Review · Reviewer_iymL · 2025-10-26

**Soundness:** 3
**Presentation:** 3
**Contribution:** 3
**Rating:** 6
**Confidence:** 4

**Summary:**

This paper addresses the poor cross-domain generalization of Process Reward Models (PRMs), which typically excel in mathematics but fail in other domains. The authors posit this is because traditional PRMs learn domain-specific correctness rather than domain-agnostic logical flow . They propose ContextPRM, a novel framework that shifts the learning objective to modeling the contextual coherence between reasoning steps. This is achieved through a synergistic combination of a new data annotation standard that labels the logical validity of step-to-step transitions and a corresponding context-aware training methodology that forces the model to evaluate these transitions in isolation . Experiments on the MMLU-Pro benchmark show that ContextPRM achieves state-of-the-art performance, particularly in non-mathematical domains.

**Strengths:**

The paper's primary strength lies in its novel and intuitive reformulation of the PRM learning objective. Shifting the focus from "is this step factually correct?" to the logical coherence is a novel approach to decouple reasoning ability from domain-specific knowledge, directly targeting the identified problem of cross-domain generalization.

This conceptual contribution is backed by robust empirical results. The model achieves substantial gains over strong baselines, including the prior SOTA (VersaPRM), in a wide array of non-mathematical domains . The claims are well-supported by extensive experiments.

Furthermore, the ablation studies are comprehensive. The finding that neither the context-based training nor the context-based labeling alone provides significant benefit, but their combination yields a dramatic performance increase which provides strong evidence for the authors' core claim of a necessary synergistic effect . The single-domain training analysis, which shows that training on "logic-intensive" domains yields better generalization than "knowledge-intensive" domains, is a compelling piece of evidence supporting the method's mechanism .

**Weaknesses:**

The ablation study reveals the full ContextPRM performs 2.2% worse than the baseline in math, suggesting that the method's trade-off for generalization is a notable loss in a domain that relies on strict, verifiable correctness rather than just contextual flow.

The method's practicality is dependent on a new, complex, and potentially costly data annotation pipeline . The entire approach relies on re-labeling the training data using an LLM verifier (gpt-4o-mini). This introduces a dependency on the quality of the verifier and reduces the method's simplicity.

**Questions:**

The context-aware training method intentionally restricts the model's input to adjacent step pairs $(S_{i-1}, S_i)$ to evaluate the "logical transition". This design seems to optimize for local logical coherence. However, a reasoning chain can fail because a step $S_i$ relies on a flawed premise established in a non-adjacent prior step (e.g., $S_{i-2}$), even if the local transition from $S_{i-1}$ to $S_i$ is perfectly logical.Does this input limitation mean ContextPRM is primarily constrained to evaluating local coherence? How does the model learn to distinguish a "locally coherent but globally flawed" step from a "globally correct" one, especially when its annotation strategy marks all steps after the first error as "incorrect"? Was a more detailed analysis conducted on the model's ability to trace errors back to their non-adjacent roots?

**Details Of Ethics Concerns:**

The paper does not appear to raise any ethical concerns. It does not involve human subjects, sensitive data, or potentially harmful applications.

---

> ### Author Response · Authors · 2025-11-21
>
> We sincerely thank the reviewer for the positive assessment and for highlighting the novelty of our context-aware training objective and the robustness of our empirical results. We address your concerns  below.
>
> > Full ContextPRM performs 2.2% worse than the baseline in math, suggesting a trade-off...
>
> We acknowledge this trade-off and have conducted a comprehensive **Error Type Analysis** (in Appendix A of the updated PDF) to explain it.
>
> **Error Distribution Mismatch:** Our analysis shows that errors in math domains are predominantly factual "Incorrectness" (e.g., calculation errors, formula misuse), which may caused by densely generated calculation and definition errors whereas non-math domains suffer primarily from "Logical Errors".
>
> **Optimization Objective:** ContextPRM is explicitly optimized for **logical coherence**. Since math failures are often "hard" calculation errors rather than "soft" logical transitions, our method's sensitivity to coherence comes at a slight cost to calculation precision.
>
> **Net Benefit:** The −2.2% math drop is strategic. We gain 6.5% in non-math domains (vs. baseline 2.2%), reflecting our emphasis on generalization rather than domain-specific overfitting. Fig. 5 (Sec. 4.4) further shows the model reaches +8.0% on math with only single domain data, confirming that mathematical capability is preserved and easily recoverable. This trade-off enables robust multi-domain performance.
>
> > Dependent on a new, complex, and costly data annotation pipeline... Dependency on the quality of the verifier.
>
> While we conducted annotation on an LLM verifier (`gpt-4o-mini`), we argue this enhances practicality rather than reducing it:
>
> **Simplicity:** We did not introduce any extra annotation step and data augmentation，only introduced a new annotation criteria in data synthesis pipeline contributed by VersaPRM. Our Context-Coherence-Based annotation method aims to shift the training target from "verifying facts" to "verifying transitions" with minimal change on training data, ensuring fair comparison against baselines, while remaining highly portable, easy to integrate, and computationally lightweight features.
>
> **Cost-Efficiency:** `gpt-4o-mini` is highly cost-effective. Applying our annotation method to the 85K training data results in 125924323(125.9M) input tokens and 7356223(7.35M) ouput tokens, which only costs **23.3 USD** according to the price of gpt-4o-mini（0.15 per 1M input tokens,0.60 per 1M output tokens）
>
> **Non-dependency on verifier:** Our ablation ("Context-label Only") showed that using these labels with *standard* training yielded minimal gains over baseline (0.84%). The massive gain (6.5% vs 2.2%) comes from our **training methodology**, suggesting our method is robust enough to extract strong signals even if the verifier is not perfect.
>
> > ContextPRM is primarily constrained to evaluating local coherence? How does it distinguish a 'locally coherent but globally flawed' step... especially when annotation marks all steps after the first error as 'incorrect'?
>
> We thank the reviewer for giving us such a critical question to improve our paper. Our model effectively handles global consistency through two mechanisms ingrained in our training design:
>
> **The Global Anchor (Question $Q$):**
> In our training format, the original question $Q$ is **always** provided as the input prefix.
> * Even if the model only sees the immediate previous step $S_{i-1}$ as local context, it simultaneously attends to $Q$.
> * If $S_i$ is locally consistent with $S_{i-1}$ but contradicts the global premises in $Q$, the model detects this inconsistency via the attention mechanism between $S_i$ and $Q$.
>
> **"Poisoned Context" Supervision:**
> Our annotation marks all steps after the first error as "incorrect" which is a deliberate design feature.
> * Consider a chain where $S_{i-1}$ is a "flawed premise" (globally wrong relative to $Q$).
> * During training, we feed the pair ($Q, S_{i-1}, S_i$) with a **negative label (0)**.
> * **Learning Objective:** This forces the model to learn that **"a logical continuation of a flawed premise is still a failure."** The model effectively learns to detect when the *context itself* ($S_{i-1}$) has drifted from the correct path anchored by $Q$.
>
> **Empirical Evidence:**
> Our "Fixed Set" analysis (in updated Error Type Analysis) shows that ContextPRM successfully corrects a high volume of **"Misinterpretation"** errors, which is typically a global error where the reasoning diverges from the question's intent. The model's ability to fix these proves it is not constrained to local checks but is actively maintaining global coherence with the Question $Q$.
>
> **Final Note:** We sincerely appreciate your insightful comments regarding the math–generalization trade-off, the practicality of our annotation pipeline, and the question of local versus global coherence. We hope our expanded analyses and clarifications address these concerns, and we kindly invite you to consider raising your score.

---

### Official Review · Reviewer_jNz1 · 2025-10-31

**Soundness:** 3
**Presentation:** 3
**Contribution:** 2
**Rating:** 4
**Confidence:** 3

**Summary:**

Standard Process Reward Models (PRMs) excel at mathematical reasoning but fail to generalize to non-mathematical domains due to a reliance on domain-specific knowledge. This paper introduces ContextPRM, a novel approach that addresses this limitation by shifting the learning objective from verifying isolated correctness to modeling domain-agnostic logical flow. Through a new data annotation standard and a context-aware training method focused on the logical coherence between reasoning steps, ContextPRM achieves state-of-the-art multi-domain generalization. It demonstrates a 6.5% average accuracy improvement over the majority voting baseline across nine non-mathematical MMLU-Pro domains, significantly surpassing the 2.2% gain from the previous best model, VersaPRM.

**Strengths:**

1. The paper clearly identifies a significant limitation of current Process Reward Models (PRMs): while effective in mathematical domains, they fail to generalize to non-mathematical domains like law, history, or philosophy.

1. Instead of adding more domain-specific knowledge, the paper proposes a novel shift in the learning objective: moving from verifying "domain-specific correctness" to modeling "domain-agnostic logical flow", which is an intuitive and reasonable idea.

1. In these domains, ContextPRM improves average accuracy by 6.5% over the majority voting baseline, outperforming the previous SOTA (VersaPRM), which achieved a 2.2% gain.

**Weaknesses:**

1. The model's strong cross-domain generalization comes at a cost. The ablation study reveals that the full ContextPRM model shows a 2.2% performance decrease in the math domain compared to the baseline, suggesting a trade-off was made to achieve broader applicability.

1. The qualitative "Failure Cases" reveal that the method still fails when complex, specialized knowledge is required. In examples from law (tort liability) and psychology (Rancho Los Amigos Scale), both ContextPRM and the baseline model failed to identify the fundamental initial error. This suggests that focusing on "logical flow" cannot fully compensate for a lack of deep, domain-specific understanding.

1. Overall, the contribution is marginal. To my understanding, this paper just extends the definition of "bad" steps during annotation, and introduces a simple concatenation of steps as a new "contextualized representation", which does not really tackle the challenge of PRMs.

**Questions:**

1. Why does ContextPRM underperform VersaPRM in the math domain? Could you further explain it?

1. In Section 3.1, what is the formatting function $\mathcal{F}$, just simple concatenation? How is it defined? What is the definition of the new context $\tilde{\mathcal{S}}$ and the contextualized step $\mathcal{P}$? How does the coherence label $c_i$ come? Could you further explain them?

---

> ### Author Response · Authors · 2025-11-21
>
> We appreciate the reviewer's constructive feedback,recognizing the soundness of our training goal and the impressive empirical performance of our method. We address your concerns below.
>
> > Cross-domain generalization comes at a cost... Why ContextPRM underperform in math domain?
>
> We acknowledge the slight performance dip (-2.2%) compared to VersaPRM in the math domain. To understand the root cause, we conducted a comprehensive **Error Type Analysis** (in Appendix A of the updated PDF). The results provide a clear explanation:
>
> **Different Error Distributions:** Our analysis reveals that errors in math-adjacent domains are predominantly factual **"Incorrectness"** (e.g., calculation errors, formula misuse), which constitute over 80% of errors in Math, which may caused by densely generated calculation and definition errors. In contrast, **logical errors** (e.g., logical fallacies, inconsistencies, non-sequiturs) are significantly more prevalent in non-math domains (e.g., Philosophy, Law).
>
> **Objective Mismatch for Math:** ContextPRM is explicitly optimized to detect **logical coherence** rather than factual correctness. Since math reasoning failures are often driven by "hard" factual/calculation errors rather than "soft" logical transitions, our method's advantage is less pronounced here. The slight drop suggests that focusing on coherence might trade off a small degree of sensitivity to pure calculation errors.
>
> **Acceptable Trade-off:** The −2.2% math drop is strategic. We gain 6.5% in non-math domains (vs. baseline 2.2%), reflecting our emphasis on generalization rather than domain-specific overfitting. Fig. 5 (Sec. 4.4) further shows the model reaches +8.0% on math with only single domain data, confirming that mathematical capability is preserved and easily recoverable. This trade-off enables robust multi-domain performance.
>
> > Still fails when specialized knowledge is required... focusing on 'logical flow' cannot fully compensate for a lack of deep, domain-specific understanding.
>
>
> We fully agree that logical coherence cannot hallucinate missing domain knowledge. A PRM cannot verify a fact it does not know. However, our **Correlation Analysis** in in Appendix A.2 demonstrates that performance gains are highly correlated with the prevalence of logical errors in a domain.
>
> This indicates that while we cannot fix "knowledge gaps" that actually requires mounts of training data for a model to acheive，ContextPRM is highly effective at fixing **reasoning gaps** that prevents the model from making invalid deductions *based on* the knowledge it has. The "Fixed Set" analysis confirms that our model predominantly rescues cases suffering from **Logical Fallacies** and **Misinterpretations**, which are the primary bottlenecks in complex, non-mathematical reasoning.
>
> > Marginal contribution... simple concatenation... What is the formatting function / new context / coherence label?
>
> We respectfully disagree that the contribution is marginal. The shift from "verifying facts" (cumulative) to "verifying transitions" (context-aware) ，which has been proved in our **Correlation Analysis** (Appendix A.2) is a fundamental paradigm shift that solves the issue of PRMs failing in non-math domains, as we kept the labeling method and concatenation minimal to ensure fair comparison against baselines.
>
> **Clarification on Method Details:**
>
> * **Formatting Function $\mathcal{F}$:**  It is a structured transformation that explicitly separates the premise from the deduction to turn the verification task into a local coherence check which actually works as code below:
> F(C, S) = "[CONTEXT]: C \n\n"[CURRENT STEP]: S"
>
> * **New Context $\tilde{S}_i$:** For a step $S_i$, the context is the **immediately preceding step** $S_{i-1}$ (or Question $Q$ if $i=1$). This isolates the local logical transition.
> * **Contextualized Step $P_i$:** This is the output of $\mathcal{F}(\tilde{S}_i, S_i)$. It represents the "atomic unit" of logical reasoning that our model evaluates.
> * **Coherence Label $c_i$:** This binary label ($1$ for coherent, $0$ for incoherent) is derived from our **Context-Coherence Based Annotation** (Section 3.3). Unlike traditional labels that only check if a step is "factually correct" w.r.t the final answer, $c_i$ also checks: "Does $S_i$ follow logically from $S_{i-1}$?" Even if $S_{i-1}$ and $S_i$ are factually correct, they can be misleading and harmful for solution. This disentangles logic from truth, which is the core innovation enabling our generalization.
>
> We hope this clarifies that our method is a targeted architectural and objective shift designed to capture the universal structure of reasoning.
>
> **Final Note:**
> Thank you for the thoughtful feedback regarding the math–generalization trade-off, the nature of our contribution, and the clarity of our methodological components. We hope our expanded analyses and detailed clarifications have adequately addressed your concerns and kindly invite you to consider raising your score.

---

### Official Review · Reviewer_wDYj · 2025-11-02

**Soundness:** 3
**Presentation:** 3
**Contribution:** 3
**Rating:** 6
**Confidence:** 3

**Summary:**

The paper suggests context-coherence-based process reward modeling. Unlike past works that aim to evaluate the isolated correctness of a specific step, this work aims to assess its logical connection with the previous step. In some sense, it sounds a bit like a NLI task, aiming to evaluate whether two connected segments are entailments or contradictions.

The paper creates a labeled dataset via gpt-4o-mini and trains Llama-3.1-8B-PRM800K on the dataset. The performance of the trained model is showcased by a labeled subset of MMLU-Pro. Based on the performance gains achieved when the model is used as a external verifier.

The trained model outperform baslines highlighting the effectivenss of the proposed method.

**Strengths:**

(1) The paper proposes an NLI-like training schema for PRMs

**Weaknesses:**

(1) I might have missed something, but I am not yet persuaded that this is a big difference from what other PRMs have been doing. Other PRM models also receive the S-1 steps (saying step S is the one of interest). While they may have not been explicility trained to look at logical connection of previous steps with step S I think that would have been kept into account. Accordingly in Figure 3 we can see that the performance gains is more pronounced in non-math domains. Which I think is not surprising as the authors trained on additional non-math data.

(2) I think the paper would have been more interesting if their was more focus on math. Math relies heavily on logical coherence of steps, which I think is a more suitable domain for the proposed method.

(3) A qualitative study on whether the trained model effectively detects different types of errors, compared to correctness-only PRMs, might have helped persuading me about (1), that this model is different from past models

(4) Not a major issue, but I would have preferred if the training have been done to different models as well. Is the PRM800k necessary for this to work? Does it work in smaller/bigger models?

**Questions:**

See weaknesses.

---

> ### Author Response · Authors · 2025-11-21
>
> We appreciate the reviewer for the thoughtful feedback, recognizing the **(i) novelty of our NLI-like training schema** and the **(ii) effectiveness of the proposed method**. We are also grateful for the insightful advice. We have addressed your specific concerns below.
>
> > Other PRM models also receive the S-1 steps (saying step S is the one of interest).
>
>
> While legacy PRMs indeed receive previous steps, they typically treat history merely as a **passive prefix**. As the context length grows, these models often fail to isolate the root cause of errors, struggling to capture the specific contextual logic relevant to the current step. In contrast, our method explicitly restructures the input to force the model to **evaluate the logical transition** between the context and the current step as the primary training objective. This ensures the model captures the relevant logical dependencies rather than simply attending to the history sequence.
>
> >  "The performance gains is more pronounced in non-math domains. Which I think is not surprising as the authors trained on additional non-math data."
>
>
> We respectfully wish to clarify a misunderstanding: **ContextPRM was trained on the exact same non-math data source as the baseline (VersaPRM)**; no new or additional data was introduced.
> The performance disparity stems entirely from our method's superior **data efficiency**, not data volume.
> * **Efficiency:** Despite using identical data, ContextPRM achieved a **6.5%** gain in non-math domains—nearly triple the baseline's gain (2.2%).
> * **Ablation Evidence:** Our ablation study demonstrates that using our re-annotated labels with the standard training method (used by VersaPRM) yields only a marginal **0.84%** improvement. This conclusively proves that our **Context-Aware Training methodology** is the true driver of performance, enabling the model to learn significantly more from the same amount of data.
>
> > I think the paper would have been more interesting if there was more focus on math... A qualitative study on whether the trained model effectively detects different types of errors... might have helped persuading me.
>
>
> We appreciate this insightful suggestion. To address this, we conducted a comprehensive **Error Type Analysis** (in Appendix A of the updated PDF) comparing ContextPRM with the baseline. The results offer a compelling explanation for the model's distinct behavior:
>
> 1.  **Nature of Errors (Math vs. Humanities):** Our analysis reveals that errors in math-adjacent domains are predominantly factual **"Incorrectness"** (e.g., calculation errors, formula misuse), often caused by dense generation errors inherent to the generator model. In contrast, **logical errors** (e.g., fallacies, inconsistencies, non-sequiturs) are significantly more prevalent in non-math domains.
> 2.  **ContextPRM Targets Logical Flaws:** Comparing failure cases, ContextPRM significantly reduces the proportion of logical errors compared to the baseline, directly leading to a reduction in total errors.
> 3.  **Mechanism of Improvement:** We found a **strong positive correlation ($r=0.80$)** between the ratio of logical errors in a domain and the performance improvement achieved by ContextPRM.
>
> This confirms that **ContextPRM realizes its improvements specifically by correcting logical flaws and enhancing contextual coherence**, rather than merely refining factual accuracy. This unique capability makes it particularly effective in non-math domains where reasoning relies more on logical flow than on calculation, distinguishing it fundamentally from correctness-only PRMs.
> >  Is the PRM800k necessary for this to work? Does it work in smaller/bigger models?
>
>
> We chose `Llama-3.1-8B-Instruct` pre-trained on `PRM800K` as our base model specifically to ensure a **fair and controlled comparison** with **VersaPRM**, the current state-of-the-art multi-domain baseline, which released this specific backbone. To strictly isolate the contribution of our method, we performed LoRA fine-tuning on this base using the exact same 85k sample size as the baseline.
> While we agree that pre-training with richer data or exploring scaling laws on different model sizes could yield further improvements, reproducing the full pre-training stage (800K samples) vs. our efficient fine-tuning (85K samples) was beyond our current computational constraints. Nevertheless, we prioritized rigorous validation through extensive ablation studies and error type analyses under these constraints to robustly demonstrate the effectiveness of our proposed methodology.
>
> **Final Note:**
> We sincerely appreciate your thoughtful feedback, especially regarding the novelty of our NLI-style training, the source of performance gains, and the domain-specific behavior of error detection. We hope our detailed clarifications and added analyses address your concerns, and we kindly invite you to consider raising your evaluation.

---

### Author Response · Authors · 2025-12-02

We thank the Area Chair and reviewers for their time and insightful feedback. We are encouraged that **all reviewers recognized the novelty** of our NLI-like training schema and the **effectiveness** of ContextPRM in achieving state-of-the-art performance in multi-domain reasoning.

During the rebuttal, we addressed the primary concerns regarding data efficiency, the trade-off in mathematical domains, and the underlying mechanism of our model. We summarize our key responses and additional analyses below:

### 1. Superior Data Efficiency (Addressing Common Concerns on Data Usage)
Reviewers questioned whether our performance gains stemmed from "additional data." We have clarified that **ContextPRM was trained on the exact same non-math data source as the baseline (VersaPRM)**; no new raw samples were introduced.
* **Efficiency:** Despite using identical data, ContextPRM achieved a **6.5%** accuracy gain in non-math domains with nearly **triple** the baseline's gain (2.2%).
* **Driver of Performance:** Our ablation studies confirm that simply using our labels with the standard training method yields minimal gains (0.84%). The substantial improvement is driven entirely by our **Context-Aware Training methodology**, which forces the model to model logical transitions rather than passive history.

### 2. Mechanism of Success: SOTA Performance Driven by Logical Coherence
Reviewers inquired about the model's behavior in different domains. We conducted a comprehensive **Error Type Analysis** (Appendix A) to demonstrate the robustness of our approach:
* **Driven by Logic:** We identified a **strong positive correlation ($r=0.80$)** between the prevalence of logical errors in a domain and our model's performance gain. This empirically proves that ContextPRM achieves State-of-the-Art (SOTA) multi-domain performance specifically by modeling logical transitions and correcting logical flaws.
* **SOTA Capability in Math:** Crucially, our method is not inherently limited in mathematics. As shown in our **One-Domain Ablation Study** (Figure 5, Sec. 4.4), when ContextPRM is fine-tuned exclusively on the math domain subset (using only ~10% of the VersaPRM training data), it achieves a remarkable **8.0% accuracy improvement**. This confirms that our Context-Aware Training framework is highly data-efficient and capable of delivering SOTA performance in the mathematics domain as well.

### 3. Clarification on Mechanism and Practicality
* **Logical Correction:** Our new analysis reveals a **strong positive correlation ($r=0.80$)** between the prevalence of logical errors in a domain and our model's performance gain, empirically proving that ContextPRM works by correcting logical flaws.
* **Global Consistency:** We clarified that our input formatting (anchoring with Question $Q$) ensures the model evaluates local transitions without losing global context, effectively fixing "Misinterpretation" errors.
* **Low Cost:** We highlighted that our annotation pipeline is highly cost-efficient (~$23 USD for 85k samples via gpt-4o-mini) and scalable, removing the bottleneck of human expert annotation.

**Conclusion:**
ContextPRM presents a fundamental shift in PRM from verifying domain-specific facts to modeling domain-agnostic logical flow. This paradigm shift not only achieves state-of-the-art results but also offers a scalable path toward general-purpose reasoning verifiers. We hope our detailed responses and new experimental evidence warrant a positive recommendation.

---

### Meta-Review · Area_Chair_wxyR · 2026-01-07

**Summary:**

The paper proposes a novel PRM framework designed to model domain-agnostic logical flow. It enables PRM to enhance generalization performance across diverse domains, going beyond domain-specific performance such as math. The reviewers acknowledge the novelty and contribution of contextPRM, which is supported by a clear problem definition, intuitive ideas, and robust empirical validation. But, they raise concerns about performance degradation in the math benchmark and heavy dependence on the LLM verifier. Additionally, it is noted that contextPRM is a reasonable initial approach rather than a comprehensive solution to the core problem.

I agree that contextPRM has lots of limitations. As the reviewers mentioned, there is a trade-off between math and general performance, suggesting that the gains in generalization do not come without a cost.  Also, its effects on complex knowledge are still doubtful. However, I believe it makes a contribution as a simple and early trial to address the generalization problem of PRM.
Overall, the reviewers' opinion leans toward acceptance. Thus, I recommend acceptance.

**Reviewer Concerns:**

Resolved concerns
- Reviewer wDYj
  - Lack of Qualitative Analysis
- Reviewer jNz1
- Reviewer iymL

Remaining concerns
- Reviewer wDYj
  - Unclear novelty
- Reviewer jNz1
  - Performance decrease in math
  - Limited effects on complex knowledge
  - Limited contribution: it doesn't really tackle the core problem
- Reviewer iymL
  - Performance decrease in math
  - The method heavily relies on LLM verifier

**Reviewer Scores:**

- Reviewer wDYj: Would maintain the current score.
- Reviewer jNz1: Would maintain the current score.
- Reviewer iymL: Would maintain the current score.

---

### Decision · Program_Chairs · 2026-01-26

Accept (Poster)